# SET-9 and SET-26 are H3K4me3 readers and play critical roles in germline development and longevity

Wenke Wang[1], Amaresh Chaturbedi[1], Minghui Wang[2], Serim An[1], Satheeja Santhi Velayudhan[1], Siu Sylvia Lee[1]*

[1]Department of Molecular Biology and Genetics, Cornell University, Ithaca, United States; [2]Computational Biology Service Unit, Cornell University, Ithaca, United States

**Abstract** *C. elegans* SET-9 and SET-26 are highly homologous paralogs that share redundant functions in germline development, but SET-26 alone plays a key role in longevity and heat stress response. Whereas SET-26 is broadly expressed, SET-9 is only detectable in the germline, which likely accounts for their different biological roles. SET-9 and SET-26 bind to H3K4me3 with adjacent acetylation marks in vitro and in vivo. In the soma, SET-26 acts through DAF-16 to modulate longevity. In the germline, SET-9 and SET-26 restrict H3K4me3 domains around SET-9 and SET-26 binding sites, and regulate the expression of specific target genes, with critical consequence on germline development. SET-9 and SET-26 are highly conserved and our findings provide new insights into the functions of these H3K4me3 readers in germline development and longevity.
DOI: https://doi.org/10.7554/eLife.34970.001

## Introduction

Dynamic regulations of histone methylation status have been linked to many biological processes (*Santos and Dean, 2004*). Recent studies have revealed that specific histone methyltransferases and demethylases can play key roles in regulating germline functions and/or modulating longevity (*Han and Brunet, 2012*; *Greer and Shi, 2012*). In *C. elegans*, loss of the COMPASS complex, which is critical for methylating histone H3 lysine 4, results in a global decrease in H3K4 trimethylation (H3K4me3) levels, and reduced brood size (*Li and Kelly, 2011*; *Robert et al., 2014*) and extended lifespan (*Greer et al., 2010*) phenotypes. Interestingly, deletion of *spr-5* or *rbr-2*, both of which encode demethylases that erase methyl marks on H3K4, also results in fertility defects (*Alvares et al., 2014*; *Katz et al., 2009*) and altered lifespan (*Alvares et al., 2014*; *Greer et al., 2010*). These findings suggest that histone methylations need to be precisely controlled to maintain longevity and germline function. It is important to note that the molecular mechanisms whereby these H3K4 modifying enzymes effect reproduction and longevity functions, including the genomic regions that they act on and how the altered H3K4 methylation levels contribute to the biological outcomes in these mutants, are largely unknown.

SET (Su(var)3–9, Enhancer of Zeste, Trithorax) domain-containing proteins represent a major group of histone methyltransferases (*Dillon et al., 2005*). We previously carried out a targeted RNAi screen to identify the SET-domain containing proteins that play a role in longevity in *C. elegans*. We found that RNAi knockdown of *set-9* and *set-26*, two closely related genes, results in significant life-span extension (*Ni et al., 2012*). SET-9 and SET-26 share 96% sequence identity and both proteins contain highly conserved PHD and SET domains. PHD domains are known to bind to specific histone modifications (*Shi et al., 2007*; *2006*), suggesting that SET-9 and SET-26 could be recruited to chromatin via binding to specific histone marks. Interestingly, the SET domain of SET-9 and SET-26

*For correspondence:
sylvia.lee@cornell.edu

**Competing interests:** The authors declare that no competing interests exist.

**eLife digest** Cells keep their DNA organized by wrapping it around groups of proteins called histones. These structures not only keep the genetic code tidy, they also affect how and when a cell uses its genes. This is because small chemical groups that are added to histones, such as a methyl group added to the fourth position of histone H3 (known as H3K4me3), affect which proteins can access the surrounding genes. This in turn determines whether those genes are likely to be on or off.

Many proteins help to regulate histone modifications, including proteins that add or remove the specific chemical groups. Enzymes that add a methyl group to histone usually contain a region called SET; while proteins containing a structure called a PHD finger can recognize histone modifications and help to amplify the signal to switch a gene on or off. SET-9 and SET-26 are two proteins containing both SET regions and PHD fingers. Found in the worm *Caenorhabditis elegans*, these proteins are 97% identical. Changes in histone modifications can affect the lifespan of these worms, and the number of offspring they produce. Recent work revealed that loss of SET-9 and SET-26 makes the worms live longer.

Now, Wang et al. use gene editing to better understand how these proteins have their effects. Experiments with worms lacking the gene for SET-9 or SET-26 or both revealed that, despite looking almost identical, SET-9 and SET-26 have different roles. Every cell in the worm makes SET-26 protein and getting rid of it increases their lifespan by affecting the activity of a protein called DAF-16. But, only the cells in the reproductive system make SET-9, and both proteins play a role in fertility.

A technique called ChIP-seq revealed where each protein attached to the genome. The PHD fingers of SET-9 and SET-26 bound to around half of the possible H3K4me3 modification sites. Not all the possible sites actually had a methyl group attached, and the pattern of binding matched the pattern of modifications. This indicates that the two proteins arrive only once the positions already have their methyl groups.

Getting rid of the SET-9 and SET-26 proteins increased the number of H3K4me3 sites with methyl groups attached. This suggests that the role of SET-9 and SET-26 is to stop the spread of H3K4me3 modifications, controlling the use of certain genes.

In mammals, the proteins SETD5 and MLL5 likely do the job of SET-9 and SET-26. Understanding how they work in worms could further our understanding of fertility and ageing in humans.

DOI: https://doi.org/10.7554/eLife.34970.002

contains mutations in conserved residues thought to be key for methylating activities (*Ni et al., 2012*), making it unclear whether SET-9 and SET-26 could be active enzymes. Nevertheless, a recent study reported that the SET domain of SET-26 exhibits H3K9me3 activity in vitro (*Greer et al., 2014*).

In this work, we demonstrated that, despite their high sequence identity, SET-26, but not SET-9, plays a key role in heat stress response and longevity. In addition, we revealed a novel redundant function of SET-9 and SET-26 in germline development. We also confirmed that SET-26 is broadly expressed, whereas SET-9 is only expressed in the germline, which likely accounts for their distinct and redundant functions in lifespan and reproduction. Indeed, genetic and transcriptomic analyses supported the notion that SET-26 acts through the FOXO transcription factor DAF-16 in the soma to modulate longevity. Furthermore, we showed that the PHD domains of SET-9 and SET-26 bind to H3K4me3 in vitro and that the genome-wide binding patterns of SET-9 and SET-26 are highly concordant with that of H3K4me3 marking in *C. elegans*, indicating that SET-9 and SET-26 are recruited to H3K4me3 marked regions in vivo. Although the SET domain of SET-26 was reported to methylate H3K9me3 in vitro (*Greer et al., 2014*), our results indicated that loss of *set-9* and *set-26* does not affect global H3K9me3 levels and the genome-wide binding patterns of SET-9 and SET-26 are highly divergent from that of H3K9me3. Instead, we found that loss of *set-9* and *set-26* results in expansion of H3K4me3 marking surrounding most if not all of the SET-9 and SET-26 binding sites specifically in the germline, and significant RNA expression change of a subset of germline specific genes bound by SET-9 and SET-26. We propose that SET-9 and SET-26 are recruited to the chromatin via binding to H3K4me3, where they function to restrict H3K4me3 spreading and to regulate the expression of

specific genes, and together these activities contribute to the proper maintenance of germline development.

## Results

### *set-26*, but not *set-9*, single mutant exhibits prolonged lifespan and heightened resistance to heat stress

RNAi knockdown of the highly similar paralogs *set-26* and *set-9* were previously shown to significantly extend lifespan in *C. elegans* (*Greer et al., 2010*; *Ni et al., 2012*). However, due to their high sequence similarity, RNAi likely knocks down both *set-26* and *set-9* in those experiments. We confirmed the lifespan extension phenotype with multiple available *set-26* single mutants, but a *set-9* single mutant was not available at the time (*Ni et al., 2012*). To delineate whether *set-9*, like *set-26*, also plays a role in lifespan determination, we used CRISPR-cas9 to generate a *set-9* mutant (*Figure 1A*). The *set-9* mutant we obtained carries a mutation that causes a premature stop codon and is expected to produce a truncated SET-9 protein lacking the conserved PHD and SET domains (*Figure 1A*). We tested the lifespan phenotype of this *set-9* single mutant along with the *set-26* single and *set-9 set-26* double mutants. Consistent with previous results, the *set-26* single mutant lived longer than wild-type worms (*Figure 1B*). Surprisingly, although SET-9 and SET-26 proteins share 97% identity in protein sequence, the *set-9(rw5)* mutation did not alter lifespan in either wild-type or the *set-26* mutant background (*Figure 1B*). Similar to the lifespan phenotype, *set-26*, but not *set-9* single mutant, was more resistant to heat stress compared to wild-type worms (*Figure 1C*). These results suggested that inactivation of *set-26*, but not *set-9*, extends lifespan and improves heat resistance.

### *set-9* and *set-26* act redundantly to maintain germline function

While propagating the *set-9 set-26* double mutant, we noticed a possible fertility defect. To more thoroughly assess the roles of SET-9 and SET-26 in reproduction, we assayed the brood size of *set-9*, *set-26* single and double mutants. We found that the progeny number produced by *set-9* and *set-26* single mutants was slightly smaller compared to that of wild-type worms (*Figure 2A*). Interestingly, the homozygous *set-9 set-26* double mutant derived from heterozygous parents (first generation, i.e. F1) also exhibited a mild brood size defect, and this defect became significantly more severe in the second and later generations (F2 to F6, *Figure 2A*). Deficiency of several histone modifiers has been previously reported to exhibit a 'mortal germline' phenotype. We performed a classical mortal germline assay and found that the *set-9 set-26* double mutant indeed displayed a mortal germline phenotype (*Figure 2B*). Further detailed analyses indicated that the *set-9 set-26* double mutant exhibited a high sterile rate and a low brood size through the F2-F6 generations that we assayed (*Figure 2A* and *Figure 2—figure supplement 1A*).

Since we noted a large difference between the brood size of the *set-9 set-26* double mutant in the F1 and F2 generations and suspected a possible maternal influence, we therefore performed a series of crosses to test this possibility. We found that *set-9 set-26* double mutants derived from homozygous *set-9 set-26* hermaphrodites crossed with heterozygous fathers exhibited a significantly more severe brood size defect compared to those from heterozygous hermaphrodites crossed with homozygous mutant fathers (*Figure 2C*). In other words, heterozygous mothers, but not heterozygous fathers, helped to maintain better germline function in the progeny. These data supported the notion of a maternal contribution in germline maintenance in the *set-9 set-26* double mutant worms.

We next used DAPI staining to monitor the germ cells of the *set-9 set-26* double mutant at the F3 and F4 generations. For the double mutant worms that became sterile, we observed variable germline phenotypes, including a very small mitotic region with no differentiated cells, and a small mitotic region with sperms only or a largely normal mitotic region with oocytes only (*Figure 2—figure supplement 1B*), suggesting problems with both the germline stem cells and their subsequent differentiation. For the double mutant worms that remained fertile, we observed germlines with a smaller but stable number of mitotic cells (*Figure 2D*). The results together indicated that SET-9 and SET-26 act redundantly to maintain normal germline function and they may regulate both the proliferation and differentiation of the germline stem cells.

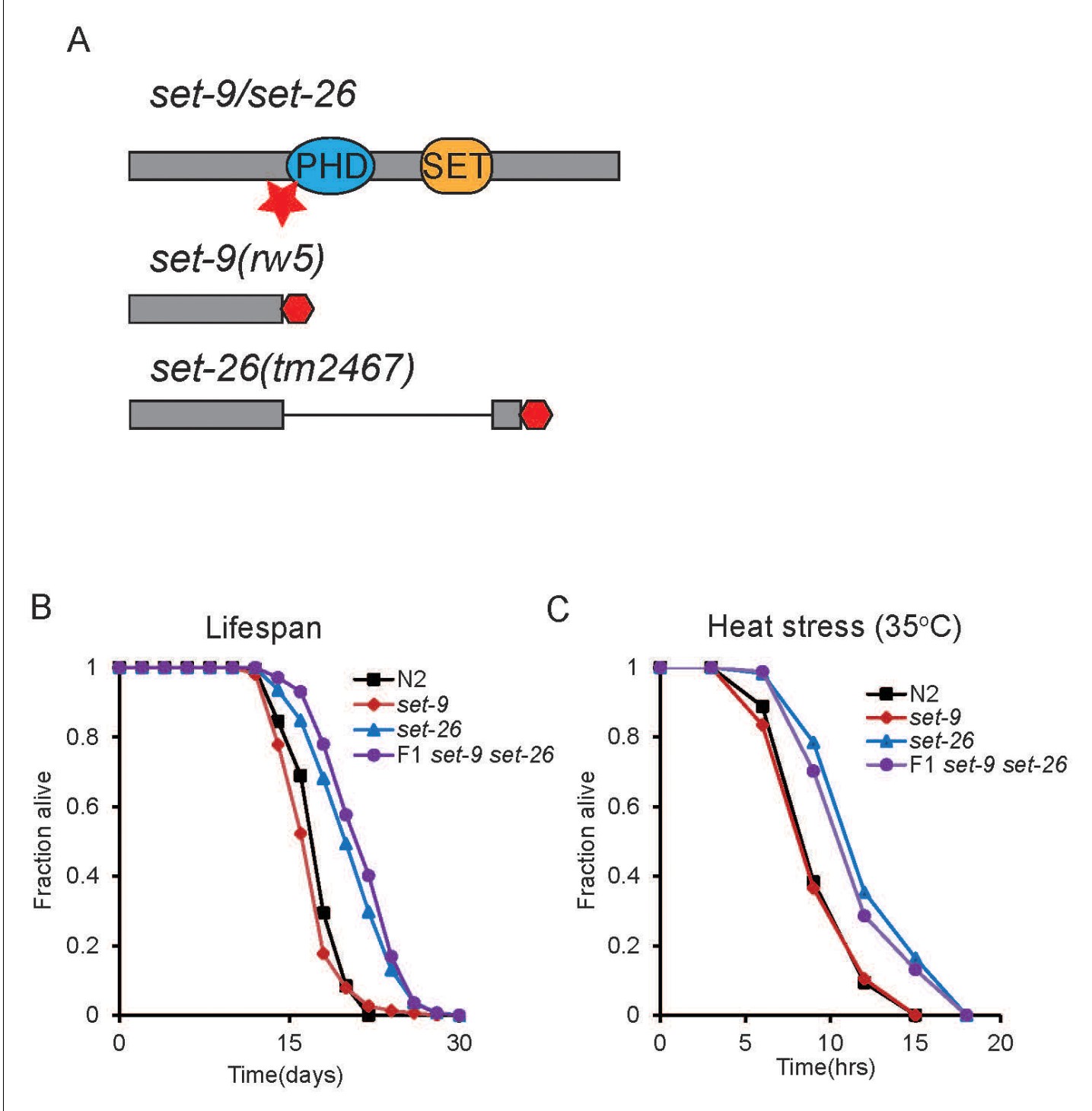

**Figure 1.** *set-26* but not *set-9* is important for longevity. (**A**) Schematic of the *set-9(rw5)* and *set-26(tm2467)* mutants. Red star indicates the position of the sgRNA (single guide RNA) targeting the *set-9* gene. Premature stop codons caused by deletions of 38 nucleotides in the *set-9* gene and 1090 nucleotides in the *set-26* gene are depicted as red hexagons. Loss of *set-26* gene but not *set-9* gene extended lifespan (**B**), and increased resistance to heat stress (**C**). Survival curves for N2, *set-26(tm2467)*, *set-9(rw5)*, and *set-9(rw5) set-26(tm2467)* strains from representative experiments are shown. Quantitative data for all replicates are shown in *Supplementary file 1* Table S1.

DOI: https://doi.org/10.7554/eLife.34970.003

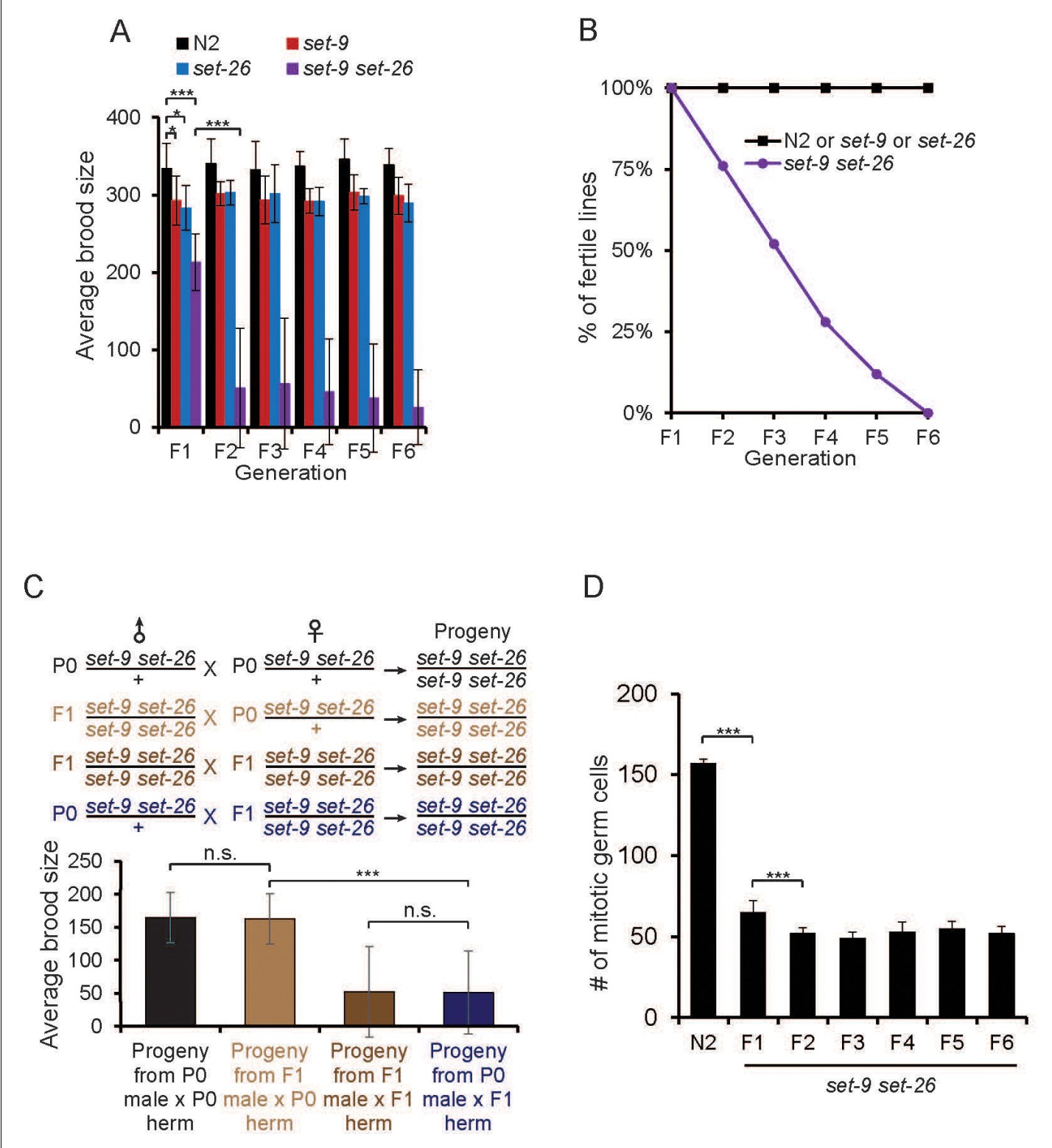

**Figure 2.** *set-9* and *set-26* act redundantly to maintain fertility. (**A**) The *set-9(rw5) set26(tm2467)* double mutant worms derived from heterozygous parents (**F1**) displayed a mild fertility defect. The double mutant worms displayed a much more severe fertility defect at later generations (F2–F6). Average brood size of N2, *set-26(tm2467)*, *set-9(rw5)*, and *set-9(rw5) set-26(tm2467)* strains at the indicated generation were shown (*p<0.05, ***p<0.001). The error bars represent standard errors. n = 9 ~10 for N2, *set-9* mutant, *set-26* mutant, and F1 *set-9 set-26* double mutant worms; n = ~50 for F2-F6 *set-9 set-26* double mutants. (**B**) The *set-9(rw5) set26(tm2467)* double mutant exhibited a mortal germline phenotype. At each generation, 6

*Figure 2 continued on next page*

*Figure 2 continued*

L1s for N2, *set-26(tm2467)*, *set-9(rw5)* and *set-9(rw5) set-26(tm2467)* strains were transferred to a new plate. Plates were scored as not fertile when no progeny were found. % of fertile lines indicated percentage of plates that were fertile. n = 6 for N2, *set-9* and *set-26* mutants; n = 25 for *set-9 set-26* double mutants. (**C**) Maternal contribution of *set-9* and *set-26* appeared important for alleviating the fertility defect in the double mutant. Average brood size of the *set-9(rw5) set26(tm2467)* double mutants derived from four different crosses were shown (***p<0.001, n.s. no significant). n = 11 ~ 12 for assessing the brood size of the *set-9(rw5) set26(tm2467)* homozygous progeny from heterozygous male(P0) X heterozygous hermaphrodite(P0) and homozygous male(F1) X heterozygous hermaphrodite(P0); n = 30 ~34 for progeny from homozygous male (F1) X heterozygous hermaphrodite(P0) and heterozygous male(P0) X homozygous hermaphrodite(F1). (**D**) The *set-9(rw5) set26(tm2467)* double mutant worms that remained fertile nevertheless exhibited reduced number of mitotic germ cells. Whole worms or dissected gonads of fertile *set-9(rw5) set26(tm2467)* mutants were stained by DAPI and the mitotic cells were counted. D2 adults were scored. n = 18 ~ 27, ***p<0.001. Analyses of sterile *set-9(rw5) set26(tm2467)* double mutant worms are shown in *Figure 2—figure supplement 1*. Quantitative data are shown in *Supplementary file 1* Table S2.

DOI: https://doi.org/10.7554/eLife.34970.004

The following figure supplement is available for figure 2:

**Figure supplement 1.** Fertility defects of the *set-9 set-26* double mutant.

DOI: https://doi.org/10.7554/eLife.34970.005

## SET-26 is broadly expressed but SET-9 is only detectable in the germline

Given the high degree of sequence identity between SET-9 and SET-26, and given their differential roles in lifespan and heat resistance, we wondered whether these two proteins could be expressed in different tissues. In an attempt to resolve the expression patterns of SET-9 and SET-26, we previously used RT-PCR at precise temperatures, as well as an antibody that recognized both SET-9 and SET-26, in wild-type, *set-26* single, and germlineless mutant worms, and deduced that SET-26 is likely broadly expressed and SET-9 is likely expressed in the germline (*Ni et al., 2012*). To unambiguously determine the expression patterns of the SET-9 and SET-26 proteins, we used CRISPR-cas9 to knock-in a GFP tag at the C-terminus of the endogenous *set-9* and *set-26* loci and monitored their expression patterns. Consistent with our previous report (*Ni et al., 2012*), we found that GFP-tagged SET-9 was only detected in germline cells of *C. elegans* (*Figure 3A*). In contrast, the GFP-tagged SET-26 was broadly expressed in both the somatic and germline cells (*Figure 3B*). As expected, expression of these two proteins was restricted to the nucleus, which is consistent with their possible roles in chromatin regulation. The ubiquitous expression of SET-26, but not SET-9, likely explains why SET-26 alone has a role in lifespan and heat resistance.

We noted that the knock-in worms expressing GFP-tagged SET-26 lived slightly longer than wild-type (but significantly shorter than the *set-26* mutant) (*Figure 3—figure supplement 1A*) and had a slight heat resistance phenotype (*Figure 3—figure supplement 1B*), and the knock-in worms expressing both SET-9::GFP and SET-26::GFP had a slightly lower brood size compared to wild-type worms, but a significantly larger brood size than the *set-9 set-26* double mutant worms (*Figure 3—figure supplement 1C*). The data together suggested that the GFP-tags somewhat compromise the functions of SET-9 and SET-26, but the tagged proteins remain largely functional.

We next wondered whether the germline or somatic expression of SET-26 is important for lifespan modulation. We previously showed that RNAi knockdown of *set-9/–26* (RNAi targets the two genes due to high sequence identity) in *glp-1(e2141)* germlineless mutant worms extended lifespan to a similar degree as in wild-type worms (*Ni et al., 2012*), suggesting that somatic *set-26* is important for lifespan modulation. To further test this possibility, we used the *rrf-1* mutant, in which RNAi is efficient in the germline but not somatic cells (*Sijen et al., 2001*), to assess whether knockdown of *set-26* (and *set-9*) in the germline alone can extend lifespan. As a control for tissue-specific RNAi, we monitored SET-26::GFP expression in wild-type or *rrf-1* mutant worms treated with *set-9/–26* RNAi. As expected, *set-9/–26* RNAi greatly reduced SET-26::GFP expression in most tissues except neurons in wild-type worms (*Figure 3—figure supplement 1D*), whereas *set-9/–26* RNAi treatment specifically knocked down SET-26::GFP expression in the germline in *rrf-1* mutant worms (*Figure 3—figure supplement 1D*). We next assessed the lifespan of wild-type or *rrf-1* mutant worms treated with *set-9/–26* RNAi. We included *wdr-5.1* RNAi as a positive control as *wdr-5.1* is known to act in the germline to modulate lifespan (*Greer et al., 2010*). As expected, RNAi knockdown of *wdr-5.1* extended lifespan in both wild-type and *rrf-1* mutant worms. In contrast, *set-9/–26* RNAi knockdown extended lifespan in wild-type but not in the *rrf-1* mutant background (*Figure 3C and D*), indicating

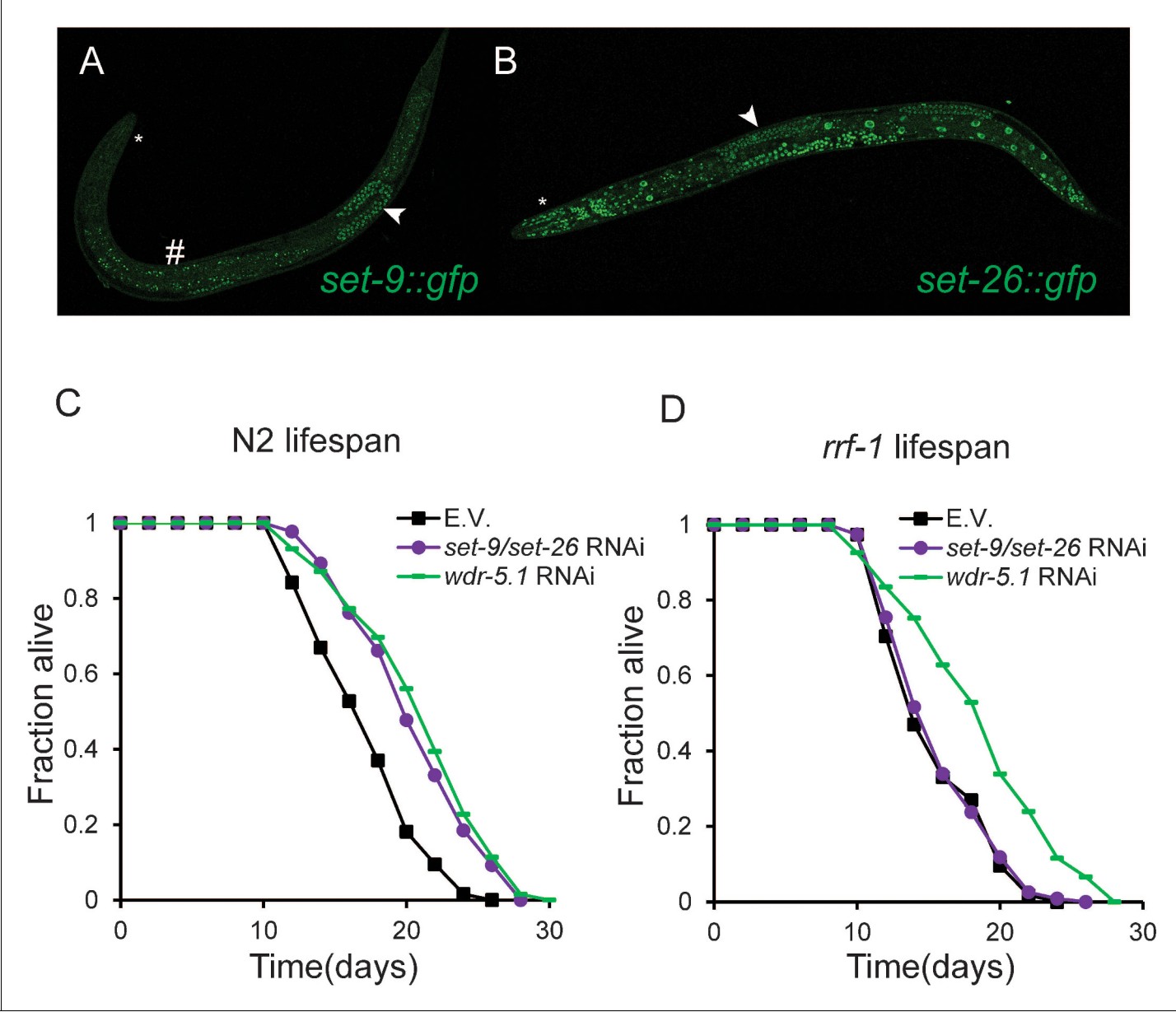

**Figure 3.** SET-26 is broadly expressed and SET-9 is only detectable in the germline. (**A, B**) Fluorescent micrographs of worms carrying *gfp* knock-in at the C-terminus of *set-9* or *set-26* gene (*set-9::gfp* and *set-26::gfp*). GFP-fused SET-26 was detected in all cells and GFP-fused SET-9 was only detected in the germline. Star indicates head, arrow indicates germline in the images. The signal outside of the germline detected in the *set-9::gfp* strain represented autofluorescence (marked by hashtag), which appeared yellow under the microscope. (**C, D**) Germline-specific knockdown of *set-9* and *set-26* was not sufficient to extend lifespan. RNAi Knockdown of *set-9 and set-26* or *wdr-5.1* extended lifespan in N2 worms (**C**). RNAi knockdown of *wdr-5.1*, but not *set-9 and set-26*, extended lifespan in the *rrf-1(pk1417)* mutant worms. Quantitative data are shown in *Supplementary file 1* Table S3.
DOI: https://doi.org/10.7554/eLife.34970.006

The following figure supplements are available for figure 3:

**Figure supplement 1.** GFP-tagged SET-9 and SET-26 are largely functional.
DOI: https://doi.org/10.7554/eLife.34970.007

**Figure supplement 2.** GFP-tagged SET-9 is only detectable in the germline.
DOI: https://doi.org/10.7554/eLife.34970.008

that inactivation of *set-26* (and *set-9*) in the germline is not sufficient for lifespan modulation. These results corroborated with our previous findings, and indicated that SET-26 likely acts in the somatic cells to modulate longevity and heat stress response, but SET-9 and SET-26 act redundantly in the germline to maintain reproductive function.

## Transcriptional profiling revealed candidate longevity and germline function genes regulated by SET-9 and SET-26

To gain insights into the molecular changes that may contribute to the somatic SET-26 effect on lifespan, we investigated the transcriptional profiles of the long-lived germlineless *glp-1; set-26* double mutant. We isolated total RNA from *glp-1; set-26* double and *glp-1* single mutant worms, and performed RNA sequencing after removing ribosomal RNAs (ribo-minus RNA-seq). We next used edgeR, an RNA-seq analysis tool in the R package (*Robinson et al., 2010*), to identify the genes that showed statistically significant expression change in the *glp-1; set-26* double mutant compared to *glp-1* mutant (*Figure 4A*). We identified 887 up-regulated and 946 down-regulated genes in response to *set-26* loss in the soma (*Figure 4A*), and gene ontology (GO) analyses indicated that these genes were over-represented in multiple functional groups (*Figure 4—figure supplement 1A*), with 'collagen' stood out as the most highly enriched GO term. We noted that collagens, as well as some of the other genes with expression change, have been implicated to be important for lifespan in *C. elegans* (*Ewald et al., 2015*). It would be interesting to test how altered expression of collagens, and other genes identified in our RNA-seq data, might contribute to the extended lifespan of the *set-26* mutant in the future.

Since we previously showed that somatic *set-26* largely acts through *daf-16*, which encodes the Forkhead box O (FOXO) transcription factor, to modulate lifespan (*Ni et al., 2012*), we sought to further identify the transcriptional changes in response to somatic *set-26* loss that are also dependent on *daf-16*. Using similar RNA-seq experiments, we investigated the transcriptional profiles of the germlineless *daf-16; glp-1; set-26* triple and *glp-1; set-26* double mutants (*Figure 4A*). We identified 164 genes that were up-regulated, and 131 genes that were down-regulated in the *daf-16; glp-1; set-26* triple mutant (*Figure 4A*). By comparing these gene lists with the gene lists discussed above for the germlineless *glp-1; set-26* double mutant vs. *glp-1*, we deduced the somatic genes whose expression become significantly up-regulated or down-regulated when *set-26* is deleted, but those expression changes were reverted when *daf-16* was simultaneously lost (down-regulated or up-regulated in the *daf-16; glp-1; set-26* triple mutant, respectively) (*Figure 4A*). We termed these DAF-16-depedent somatic SET-26 regulated genes. Interestingly, GO term analyses revealed that the functional group 'determination of adult lifespan' was highly enriched in these DAF-16-dependent somatic SET-26 regulated genes (*Figure 4B*). Therefore, the transcriptomic analysis corroborated the genetic analysis, and supported a model that DAF-16-mediated gene regulation likely contributes to the lifespan phenotype of the *set-26* mutant.

We additionally investigated the transcriptional profiles of the long-lived fertile *set-26* single mutant and revealed that 869 genes showed significant expression change in the *set-26* mutant compared with wild-type worms. As expected, there was a significant and substantial overlap between the genes that exhibited expression change in response to whole-body loss of *set-26* and the somatic SET-26 regulated genes discussed above (*Figure 4—figure supplement 1B*). Interestingly, the analysis using germlineless worms revealed far greater number of genes with expression change compared to that using reproductive worms. This could be due to technical variations between experiments, but might also suggest that some genes exhibit selective expression changes only in somatic cells, and those expression changes could be masked when germ cells were included in the analysis.

To gain insights into the molecular changes that may underlie the germline phenotypes, we next compared the transcriptional profiles of the *set-9* single, *set-26* single, and F1 *set-9 set-26* double mutants, all of which were fertile. We identified 162, 334, 1888 genes that were up-regulated, and 545, 534, 1644 genes that were down-regulated in the *set-9*, *set-26*, and F1 *set-9 set-26* mutants respectively (*Figure 5A*). Interestingly, although there was significant overlap among the three gene sets, a substantial number of genes appeared to only show expression changes in the F1 *set-9 set-26* double mutant (*Figures 5A*, 1430 down-regulated, 1781 up-regulated), suggesting a redundant role of SET-9 and SET-26 in regulating gene expression. Since the F1 *set-9 set-26* mutant had a mild brood-size phenotype, and gave rise to progeny that exhibited severe defects in germline

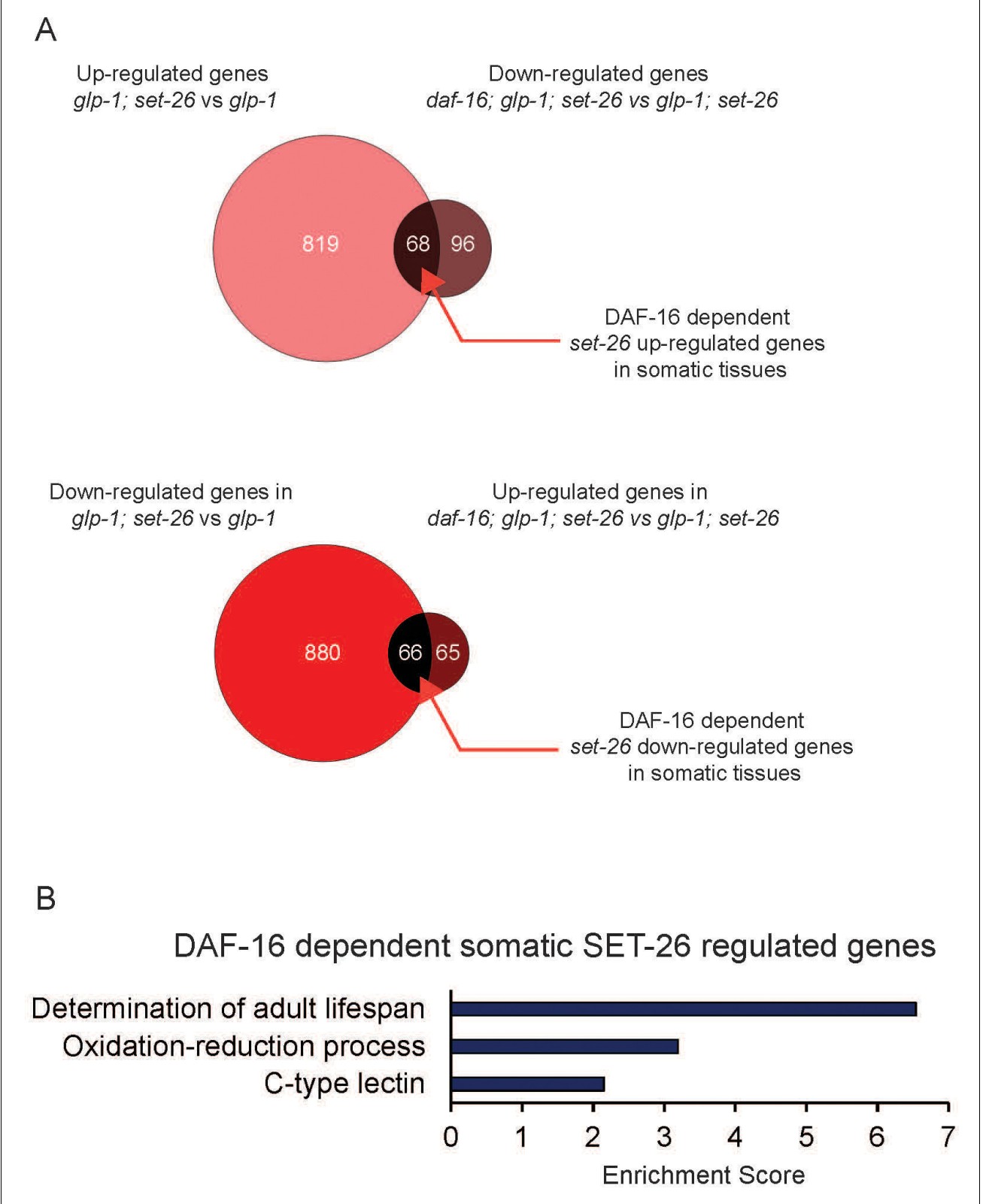

**Figure 4.** DAF-16-dependent somatic SET-26 regulated genes are enriched for lifespan determinant genes. (**A**) Venn diagrams show the overlap between up-regulated genes in *glp-1(e2141); set-26(tm2467)* (comparing with *glp-1(e2141)*) and down-regulated genes in *daf-16(mgDf47); glp-1(e2141); set-26(tm2467)* (comparing with *glp-1(e2141); set-26(tm2467)*); and the overlap between down-regulated genes in *glp-1(e2141); set-26(tm2467)*

*Figure 4 continued on next page*

*Figure 4 continued*

(comparing with *glp-1(e2141)*) and up-regulated genes in *daf-16(mgDf47); glp-1(e2141); set-26(tm2467)* (comparing with *glp-1(e2141); set-26(tm2467)*).
(B) GO term analysis of DAF-16-dependent somatic SET-26 regulated genes. Gene lists can be found in **Supplementary file 2**.
DOI: https://doi.org/10.7554/eLife.34970.009
The following figure supplement is available for figure 4:

**Figure supplement 1.** Transcriptional profile of *glp-1(e2141); set-26(tm2467)* mutant.
DOI: https://doi.org/10.7554/eLife.34970.010

development (**Figure 2**), we speculated that many of these SET-9 and SET-26 co-regulated genes could be important for germline function. Indeed, GO analyses revealed that the genes that showed expression change in response to the simultaneous loss of *set-9* and *set-26* were over-represented for a number of different functional groups, including genes with annotated functions in sperm development and function (**Figure 5B–C**). We further compared these SET-9 and SET-26 co-regulated genes with genes previously determined to be germline-, oocyte-, and sperm-specific (**Reinke et al., 2004**). Interestingly, we found a significant over-representation of germline-specific genes among the genes that exhibited up-regulated expression in the F1 *set-9 set-26* double mutant, but not the genes that exhibited down-regulated expression (**Figure 5D**). Both sperm- and oocyte-specific genes (383 and 252, respectively) were among these germline-specific genes that were up-regulated in the F1 *set-9 set-26* double mutant. It is possible that up-regulated expression of these germline-specific genes contribute to the reproductive defects of the *set-9 set-26* double mutant (**Greer et al., 2014**; **Katz et al., 2009**; **Kerr et al., 2014**).

Considering the maternal effect of SET-9 and SET-26 on fertility (**Figure 2C**), we also profiled the transcriptome of the F3 *set-9 set-26* double mutant, which exhibited greatly compromised fertility (**Figure 2A**), and compared that with the transcriptional profile of the F1 *set-9 set-26* double mutant discussed above. We noted that the germline of the F3 *set-9 set-26* was morphologically quite different from wild-type and the F1 *set-9 set-26* double mutant (**Figure 2—figure supplement 1**). Interestingly, we found that the genes with significant expression change in the F1 *set-9 set-26* and F3 *set-9 set-26* double mutants not only substantially overlapped, but they were also enriched for similar functional groups based on GO term analyses (**Figure 5—figure supplement 1A, B and C**). The fold change of gene expression, compared to wild-type, in the F1 *set-9 set-26* and the F3 *set-9 set-26* double mutants also positively correlated (**Figure 5—figure supplement 1D and E**). These results together suggested that the transcriptional profiles of the F1 and the F3 *set-9 set-26* double mutants are highly correlative despite that the F3 *set-9 set-26* double mutant has a more severely defective germline. We note that the GO term 'development and reproduction' was unique for the F3 *set-9 set-26* (**Figure 5—figure supplement 1B**), which may reflect the more severe germline defects in these mutant worms.

## Loss of SET-9 and SET-26 does not affect the global levels of H3K9me3

We next investigated the possible normal functions of SET-9 and SET-26, which could inform how their inactivations lead to the gene expression changes and biological phenotypes discussed above. The SET domain of SET-26 was recently reported to show H3K9me3 methylation activity in vitro (**Greer et al., 2014**). This was a somewhat surprising result, as the SET domain of SET-26 (and SET-9) contains multiple mutations in the critical residues generally thought to be key for the methylating enzymatic activity of SET domain proteins (**Figure 6—figure supplement 1A**), and we have previously speculated that SET-9 and SET-26 are likely not active enzymes (**Ni et al., 2012**). Furthermore, the likely homologs of SET-9 and SET-26 in flies (UpSET) and mammals (MLL5) have been reported to lack methylating activity (**Sebastian et al., 2009**; **Rincon-Arano et al., 2012**). Nevertheless, because of the reported in vitro results, we sought to monitor the global levels of H3K9me3 in the *set-9* and *set-26* mutants. We reasoned that if SET-9 and SET-26 are major enzymes for depositing H3K9me3 in *C. elegans*, then we would detect reduced H3K9me3 levels in the *set-9 set-26* double mutant strain. Using Western blotting, we showed that the global levels of H3K9me3 were not detectably altered in the *set-9 set-26* double mutants compared to wild-type worms at the L4 stage (**Figure 6—figure supplement 1B**). We further investigated the genomic distribution of H3K9me3 in wild-type N2 and the F3 *set-9 set-26* double mutant worms using ChIP-seq (chromatin

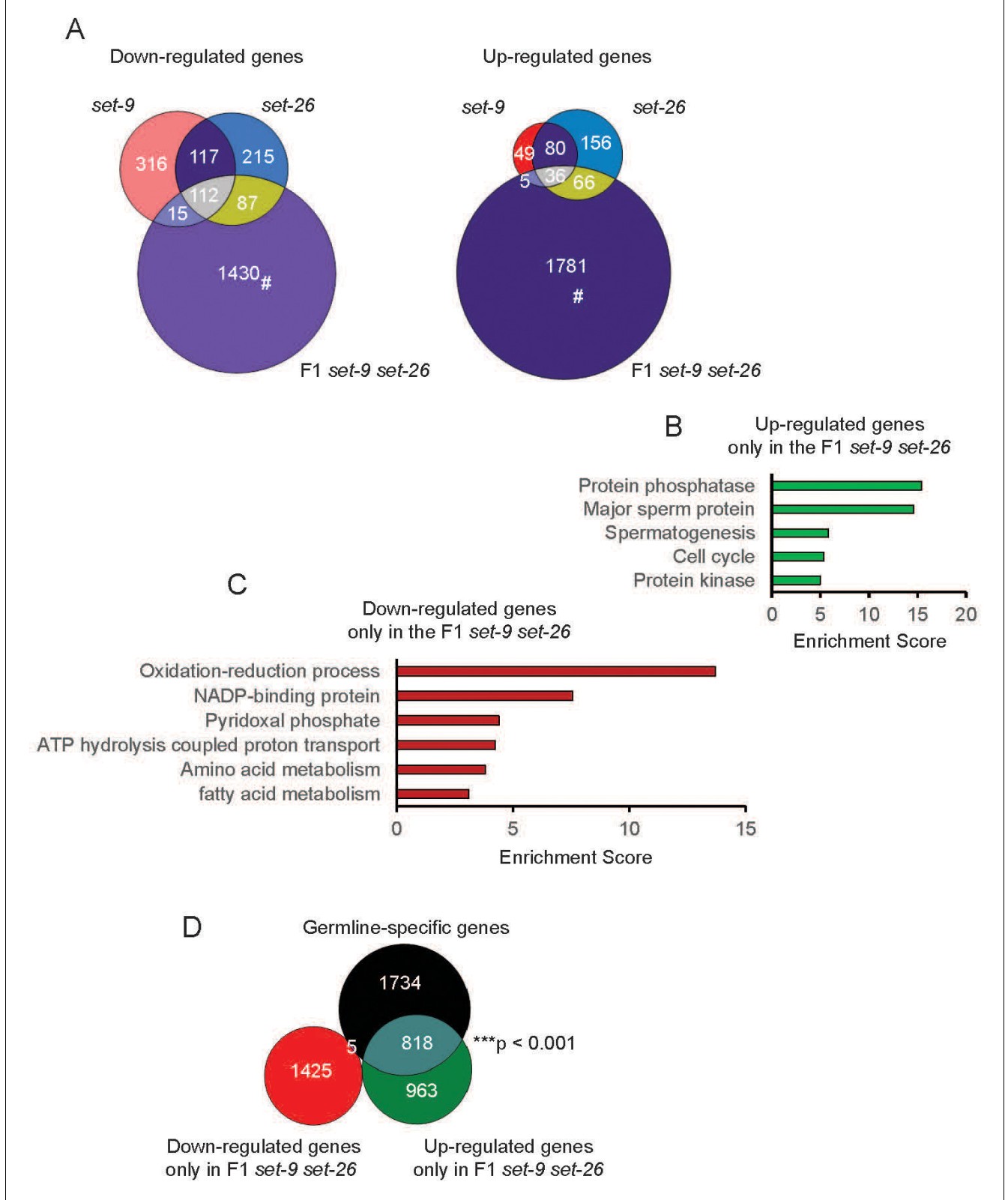

**Figure 5.** Transcriptional profiles of *set-9(rw5)*, *set-26(tm2467)*, and F1 *set-9(rw5) set-26(tm2467)* mutants. (A) Venn diagrams show the overlap among *set-9(rw5)*, *set-26(tm2467)*, and *set-9(rw5) set-26(tm2467)* down-regulated (left) and up-regulated (right) gene sets. Hashtag indicates genes that only show expression change in the F1 *set-9(rw5) set-26(tm2467)* double mutant. GO term analysis of up-regulated (B) and down-regulated (C) genes that only show expression change in the F1 *set-9(rw5) set-26(tm2467)* double mutant. (D) Venn diagram shows the overlaps between genes that only show
*Figure 5 continued on next page*

*Figure 5 continued*

expression change in the F1 *set-9(rw5) set-26(tm2467)* double mutant identified in our RNA-seq data with the previously reported germline-specific gene lists (*Reinke et al., 2004*). Gene lists can be found in *Supplementary file 2*.

DOI: https://doi.org/10.7554/eLife.34970.011

The following figure supplement is available for figure 5:

**Figure supplement 1.** Transcriptional profile comparison between F1 and F3 *set-9(rw5) set-26(tm2467)* double mutants.

DOI: https://doi.org/10.7554/eLife.34970.012

immunoprecipitation coupled with next generation sequencing). Inspection of the genome-wide H3K9me3 distribution between N2 and the F3 *set-9 set-26* double mutant revealed highly similar patterns (*Figure 6—figure supplement 1C*, *2A*). Furthermore, the Diffbind data analysis pipeline (*Bardet et al., 2011*) identified little significant differences between the two data sets (data not shown). Taken together the Western and ChIP-seq results, we concluded that SET-9 and SET-26 are likely not the major enzymes required for H3K9me3 deposition in *C. elegans*. However, we could not rule out the possibility that the SET-9 and SET-26 could deposit H3K9me3 in specific cells, or during a specific time. Indeed, we previously reported that in the germlineless mutant *glp-1*, loss of *set-26* resulted in lower levels of H3K9me3 (*Ni et al., 2012*). Further investigations will be necessary to resolve whether SET-9/SET-26 could regulate H3K9me3 levels under some circumstances. We also note that SET-25 has been shown to be important for the deposition of the majority of H3K9me3 in *C. elegans*, although the effect could be indirect as direct H3K9 methylating activities of SET-25 have yet to be demonstrated (*Towbin et al., 2012*).

## The PHD domains of SET-9 and SET-26 bind to H3K4me3 with adjacent acetylation marks in vitro

We next turned our attention to the PHD domains of SET-9 and SET-26, which are 100% identical. PHD domains are known to recognize specific histone modifications, we therefore tested whether the PHD domains of SET-9 and SET-26 also bind to specific histone modifications in vitro. We first used GST-tagged PHD domain of SET-9/SET-26 to perform an in vitro pull-down experiment using histones from calf thymus. We found that the PHD domain of SET-9 and SET-26 pulled down histones, in particular histone H3 (*Figure 6A*). A small amount of histone H4 was also recovered, which may be due to H3 and H4 associating as histone octamers in cells. We next screened for the specific histone modifications that are recognized by the PHD domain of SET-9 and SET-26 using a histone peptide array containing 95 unique modifications and 265 synthetic histone peptides. This assay revealed that the PHD domain of SET-9 and SET-26 specifically interacted with H3 peptides containing the K4me3 modification in combination with nearby acetylation (K9ac, K14ac and/or K18ac), but not H3K4me3 alone (*Figure 6B*).

## Genome-wide profiles of SET-9 and SET-26 are highly concordant with that of H3K4me3

We further examined the genome-wide binding profiles of SET-9 and SET-26 in *C. elegans* using ChIP-seq. Because antibodies capable of immunoprecipitating endogenous SET-9 and SET-26 were not available, we utilized the GFP knock-in strains discussed above. We performed anti-GFP ChIP-seq using the *set-9::gfp*, *set-26::gfp*, and *set-9::gfp set-26::gfp* strains. The double GFP strain was used in the hope that higher levels of GFP expression would provide more robust ChIP-seq data (*Figure 6—figure supplement 3*). MACS2 was used to identify the genomic regions significantly enriched for SET-9 and SET-26 binding. From the analyses, 602, 3948, and 5903 peaks were identified as bound by SET-9, SET-26, and SET-9 and SET-26 together, respectively. Interestingly, 67% of the SET-9 peaks (*Figure 6—figure supplement 3B*) overlapped with the SET-26 peaks, and 87% of the SET-26 peaks overlapped with the SET-9 and SET-26 peaks (*Figure 6—figure supplement 3C*). That SET-9 was found to bind many fewer regions than SET-26, and that the SET-9-bound peaks largely overlapped with those bound by SET-26, are consistent with the earlier data indicating SET-9 has a much more restricted expression pattern and a more limited function in the germline that is redundant with SET-26. Interestingly, ChIP-seq analysis from the double GFP strain nevertheless revealed many more peaks compared to SET-26::GFP alone. We interpreted these results to suggest

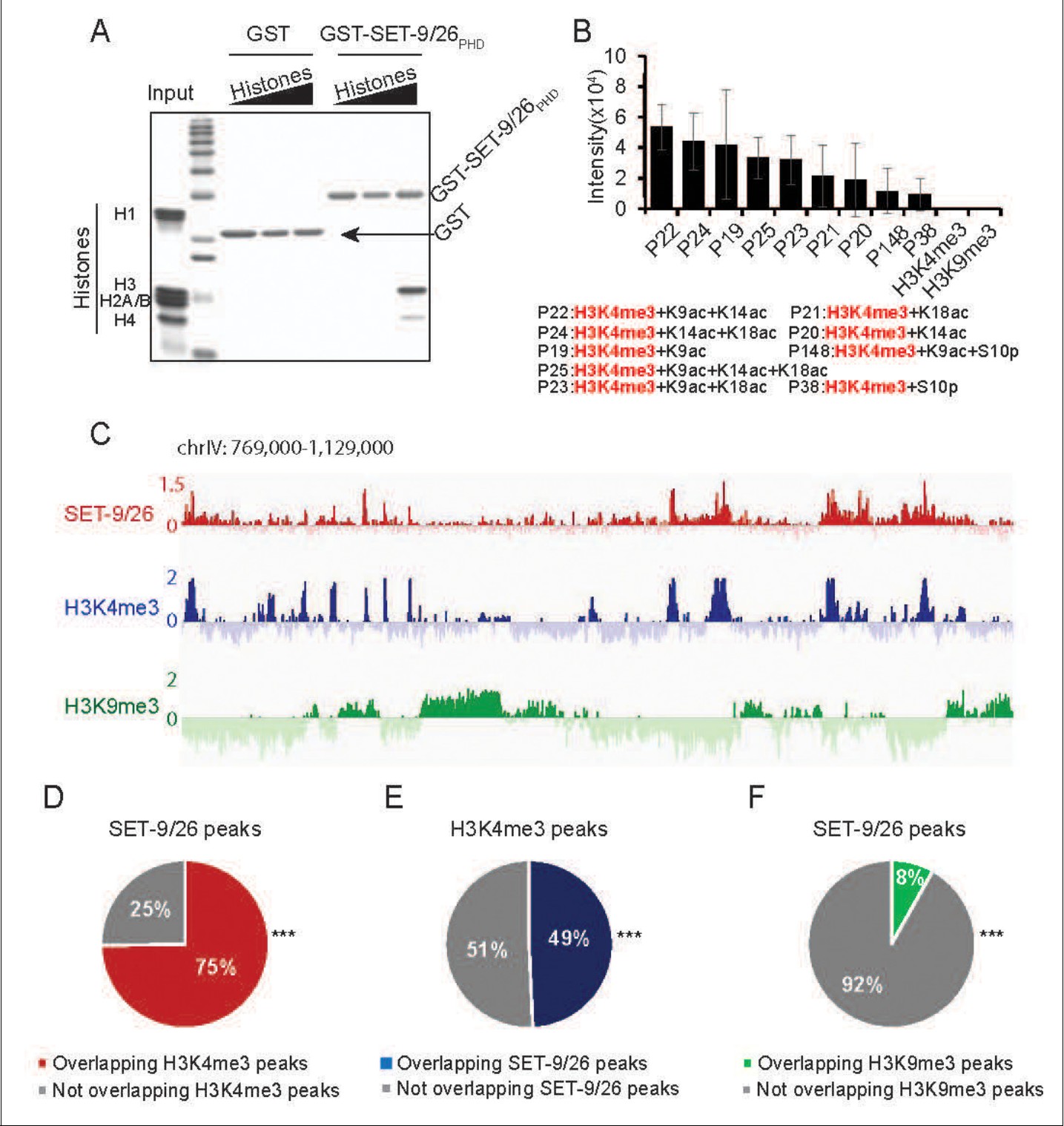

**Figure 6.** SET-9 and SET-26 bind to H3K4me3. (**A**) PHD domain of SET-9 and SET-26 specifically pulled down H3 in vitro. Coomassie-blue stained gel showing pulldown results using GST–SET-9/26$_{PHD}$ and GST control. (**B**) Binding intensity of GST-SET-9/26$_{PHD}$ to the significant hits from histone peptide arrays. Quantitative data are shown in *Supplementary file 1* Table S5. (**C**) Genome browser view showing ChIP z-scores (standardized log2 ratios of ChIP/Input or ChIP/H3ChIP signals) for SET-9/26, H3K4me3 and H3K9me3 at a representative region. (**D**) 75% of the SET-9 and SET-26 peaks overlapped with H3K4me3 peaks. (***p<0.001 indicates overlapping more than expected) (**E**) 49% of the H3K4me3 peaks overlapped with SET-9 and SET-26 peaks. (***p<0.001 indicates overlapping more than expected) (**F**) 8% of the SET-9/26 peaks overlapped with H3K9me3 peaks. (***p<0.001 indicates overlapping less than expected).

*Figure 6 continued on next page*

*Figure 6 continued*

DOI: https://doi.org/10.7554/eLife.34970.013

The following figure supplements are available for figure 6:

**Figure supplement 1.** SET-9 and SET-26 are not the major enzymes for H3K9me3 in vivo.

DOI: https://doi.org/10.7554/eLife.34970.014

**Figure supplement 2.** SET-9 and SET-26 are not the major enzymes for H3K9me3 in vivo (part 2).

DOI: https://doi.org/10.7554/eLife.34970.015

**Figure supplement 3.** SET-9 and SET-26 share similar genomic binding profiles in vivo.

DOI: https://doi.org/10.7554/eLife.34970.016

**Figure supplement 4.** SET-9 and SET-26 bind to H3K4me3 in vivo.

DOI: https://doi.org/10.7554/eLife.34970.017

**Figure supplement 5.** SET-9 and SET-26 bind to H3K4me3 in vivo (part 2).

DOI: https://doi.org/10.7554/eLife.34970.018

that the double GFP strain simply represented a better reagent for capturing the SET-9 and SET-26 binding profiles than either of the single GFP strain, because the ChIP assay worked more effectively with higher levels of GFP expression. For further analyses, we used the peak regions identified in the double GFP strain as representation of the binding sites of SET-9 and SET-26.

Given the in vitro binding of the PHD domain of SET-9 and SET-26 to H3K4me3, we wondered whether SET-9 and SET-26 also bind to H3K4me3 in vivo. To test this, we examined the genome-wide pattern of H3K4me3 in wild-type worms using ChIP-seq. A representative genome browser view revealed that the SET-9 and SET-26 binding profile generally correlated well with the H3K4me3 profile (*Figure 6C*). In contrast, the SET-9 and SET-26 binding profile was largely different from that of H3K9me3 (*Figure 6C*), again consistent with our earlier conclusion that SET-9 and SET-26 likely do not modify H3K9me3.

To more rigorously assess whether the SET-9 and SET-26 profile is concordant with the H3K4me3 profile, we tested whether SET-9 and SET-26, and H3K4me3 were enriched in overlapping genomic regions. To this end, we used MACS2 to identify 5996 genomic regions enriched for H3K4me3 marking (peaks), and then compared the degree of overlap between the SET-9 and SET-26 peaks with those of H3K4me3. In this comparison, we found that 75% of the SET-9 and SET-26 binding regions overlapped with the H3K4me3 peaks (*Figure 6D*). The reciprocal comparison revealed that 49% of the H3K4me3 peaks overlapped with the SET-9 and SET-26 binding regions (*Figure 6E*). We sought to further characterize the overlapping profiles between SET-9 and SET-26 and H3K4me3 using meta-analysis. We separated the H3K4me3 peaks into the group that bound by SET-9 and SET-26 and the group that did not, and we plotted the average normalized H3K4me3 levels centered around the summits of the H3K4me3 peaks and oriented at a 5′ to 3′ direction according the nearest genes associated with the peaks. We found that, on average, the H3K4me3 peaks bound by SET-9 and SET-26 had higher levels of H3K4me3 marking and their H3K4me3 marking was somewhat higher 5′ to the summit (*Figure 6—figure supplement 4A*). This asymmetrical marking of H3K4me3 might relate to that these peaks generally localized around annotated transcriptional/translational start sites (TSSs) and H3K4me3 levels tend to be higher 5′ to the start sites of the genes. In contrast, the average plot of the H3K4me3 peaks that were not bound by SET-9 and SET-26 was more symmetrical, and this correlated with these H3K4me3 peaks generally localized to gene body regions (data not shown).

We next used a similar meta-analysis approach but plotted the normalized SET-9 and SET-26 ChIP signals for the two groups of H3K4me3 peaks, still oriented at the summits of the H3K4me3 peaks (*Figure 6—figure supplement 4B*). This analysis allowed us to determine how far the average summit of the SET-9 and SET-26 peaks was relative to that of the H3K4me3 peaks. The results indicated that the average summit of the SET-9 and SET-26 peaks was ~100–200 bp upstream of the H3K4me3 summit (*Figure 6—figure supplement 4B*). Interestingly, for the group of H3K4me3 peaks that were not identified to share SET-9 and SET-26 enrichment based on MACS2, we nevertheless detected a small amount of SET-9 and SET-26 binding exactly at the summits of the H3K4me3 peaks (*Figure 6—figure supplement 4B*). We interpreted these results to suggest that SET-9 and SET-26 likely bind to most if not all of the H3K4me3 enriched regions, but some of the binding was too weak (or strong binding only in a subset of the *C. elegans* cells) to be called by a statistical program

like MACS2. Lastly, we used scatter plot analysis to compare the H3K4me3 enriched signal vs. the SET-9 and SET-26 binding signal for each of the H3K4me3 peak region (*Figure 6—figure supplement 4C*). The results showed that the H3K4me3 signal intensity is positively correlated with that of SET-9 and SET-26 (*Figure 6—figure supplement 4C*). The data thus far supported a model that SET-9 and SET-26 bind to regions marked by H3K4me3 in *C. elegans*, and that the detectable SET-9 and SET-26 binding sites are generally marked by higher levels of H3K4me3.

Since the in vitro histone peptide array results suggested that histone acetylations are also important for the binding, we next compared the degree of overlap between the SET-9 and SET-26 peaks with those of H3K9ac using data from modENCODE. In this comparison, we found that 28% of the SET-9 and SET-26 binding regions overlapped with the H3K9ac peaks (***p<0.001) (*Figure 6—figure supplement 5A*), and 98% of these shared peak regions were also marked by H3K4me3 (data not shown). The significant but lower percentage of overlap between SET-9 and SET-26 binding with the H3K9ac enriched regions (relative to the overlap with H3K4me3 enriched regions) is consistent with the in vitro observation that SET-9 and SET-26 bind to H3K4me3 with adjacent acetylation, but the exact acetylated residues could vary (*Figure 6B*), but is also likely partly due to technical differences, for example the different ChIP-seq data were generated using worms at different stages and acetyl specific antibodies generally have lower specificity. We additionally compared the overall correlation between H3K4me3 and H3K9ac enriched regions and whether SET-9 and SET-26 may affect their co-occurrence. As expected, we found that the genome-wide distribution of H3K4me3 correlated well with that of H3K9ac (*Figure 6—figure supplement 5B*). Interestingly, the correlation between H3K4me3 and H3K9ac was higher for the peaks that were bound by SET-9 and SET-26 compared to those that were not (*Figure 6—figure supplement 5C* and *Figure 6—figure supplement 5D*). These genomic data are consistent with the model that SET-9 and SET-26 bind to genomic regions marked by H3K4me3 with nearby acetylations in *C. elegans*.

## SET-9 and SET-26 regulate the RNA expression of some of their target genes

We next asked whether SET-9 and SET-26 binding could influence gene expression. To test this, we assigned the 5903 regions bound by SET-9 and SET-26 (based on ChIP-seq results, *Figure 6*) with their closest genes, and then filtered them for genes that were detectably expressed in our RNA-seq data sets, which yielded 4427 potential targets of SET-9 and SET-26 that were also actively expressed in our experimental conditions. We then compared the list of SET-9 and SET-26 target genes with the lists of genes that exhibited significant gene expression change in the *set-9* single, *set-26* single, and the F1 *set-9 set-26* double mutants compared to wild-type worms based on our RNA-seq data (*Figure 7A* and *Figure 7—figure supplement 1A and D*). Through this comparison, we identified the putative SET-9 and SET-26 target genes that changed expression when *set-9* and/ or *set-26* were inactivated (*Figure 7A* and *Figure 7—figure supplement 1A and D*). Out of the 735, 876, 3532 differentially expressed genes in the *set-9*, *set-26*, and F1 *set-9 set-26* double mutant strains, SET-9 and SET-26 bound to 153, 217, 641 of them respectively (*Figure 7A* and *Figure 7— figure supplement 1A and D*). GO analyses revealed some interesting over-represented functional groups among these, especially for the SET-9 and SET-26 targets that exhibited expression change in the F1 double mutant (*Figure 7B–C* and *Figure 7—figure supplement 1*).

In an attempt to identify the somatic SET-26 target genes with a role in longevity, we compared the genes bound by SET-9 and SET-26 to the genes that showed expression change in the germline-less *glp-1; set-26* double mutant (*Figure 7D*). GO analysis did not reveal functional groups that are directly linked to longevity (*Figure 7E and F*), but 'collagen' was again overrepresented. Moreover, for the DAF-16-dependent somatic SET-26 regulated genes that were highly enriched for the functional group 'determination of lifespan' (*Figure 4*), only 3 out of the 134 genes were bound by SET-9 and SET-26, a representation that is lower than expected based on random chance. Together, these data suggested that SET-26 indirectly impact DAF-16 activity and DAF-16-mediated longevity change.

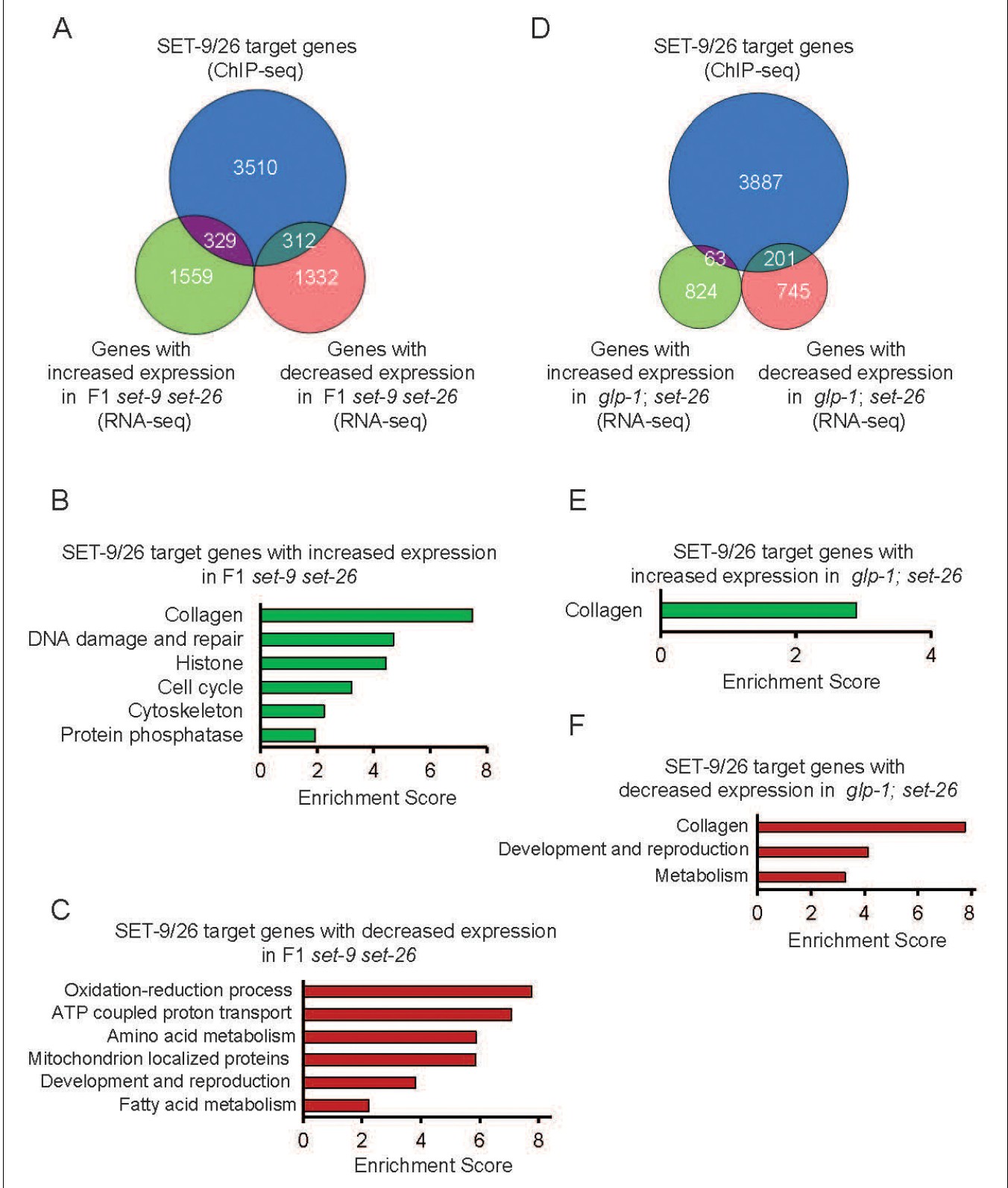

**Figure 7.** Binding of SET-9 and SET-26 regulate the RNA expression of specific target genes. (A) Venn diagram showing comparisons of the genes with increased and decreased expression changes in the F1 *set-9(rw5) set-26(tm2467)* double mutant and the SET-9 and SET-26 target genes. (B) GO term analyses of the up-regulated genes in the F1 *set-9(rw5) set-26(tm2467)* double mutant that were bound by SET-9 and SET-26. (C) GO term analyses of the down-regulated genes in the F1 *set-9(rw5) set-26(tm2467)* double mutant that were bound by SET-9 and SET-26. (D) Venn diagram showing
*Figure 7 continued on next page*

*Figure 7 continued*

comparisons of the up-regulated and down-regulated somatic SET-26 regulated genes and the SET-9 and SET-26 target genes. (E) GO term analyses of the up-regulated somatic SET-26 regulated genes bound by SET-9 and SET-26. (F) GO term analyses of the down-regulated somatic SET-26 regulated genes bound by SET-9 and SET-26. Gene lists can be found in *Supplementary file 2*.

DOI: https://doi.org/10.7554/eLife.34970.019

The following figure supplement is available for figure 7:

**Figure supplement 1.** SET-9 and SET-26 target genes.

DOI: https://doi.org/10.7554/eLife.34970.020

## Loss of SET-9 and SET-26 results in expansion of H3K4me3 marked regions

We next tested whether SET-9 and SET-26, in addition to binding to H3K4me3, can influence the marking of H3K4me3 in anyway. We first examined the global H3K4me3 levels in the *set-9* single, *set-26* single and F3 *set-9 set-26* double mutants. We observed a subtle but significant increase in H3K4me3 levels in the F3 *set-9 set-26* double mutant but not in the *set-9* or *set-26* single mutants using Western blotting (*Figure 8A* and *Figure 8—figure supplement 1A*). This increase in H3K4me3 levels was more prominent in dissected gonads of the *set-9 set-26* double mutant (*Figure 8B*).

Given our earlier data indicating that SET-9 and SET-26 bind to H3K4me3, the elevated H3K4me3 levels in the *set-9 set-26* double mutant strain could be a result of the lost recruitment of SET-9 and SET-26 at H3K4me3 sites.

To test this hypothesis, we examined the genome-wide patterns of H3K4me3 in wild-type, the F1 and F3 *set-9 set-26* double mutant worms. Inspection of the genome-wide H3K4me3 distribution in the three genotypes revealed a 'spreading' of H3K4me3 enriched regions around SET-9 and SET-26 binding sites in worms lacking *set-9* and *set-26* (*Figure 8—figure supplement 2A and B*). To quantify this 'spreading' globally, we separated the H3K4me3 enriched regions into those bound by SET-9 and SET-26 *vs.* those that were not, and compared their average profiles using meta analysis (*Figure 8C and D*). The meta analysis plots were centered around the summits of the H3K4me3 peaks and orientated in the 5' to 3' direction according to the closest gene associated with each of the peak (*Figure 8C and D*). For the H3K4me3 peaks bound by SET-9 and SET-26, we again observed a 'spreading' of H3K4me3 marking, especially towards the 3' direction. Statistically significant elevated levels of H3K4me3 were detected starting at around +500 and −1000 bp beyond the summit in the F1 and F3 *set-9 set-26* double mutant compared to wild-type worms, where the 95% confidence intervals of H3K4me3 marking in the F1 and F3 *set-9 set-26* mutants did not overlap with that in the wild-type worms (*Figure 8C*). Interestingly, for the H3K4me3 peaks that showed no detectable SET-9 and SET-26 binding, the average plot showed no significant 'spreading' of the H3K4me3 marking (*Figure 8D*). Nevertheless, the genome-wide heatmaps revealed some slight 'spreading' of H3K4me3 signals in the *set-9 set-26* double mutant even in regions not determined to be bound by SET-9 and SET-26 (*Figure 8—figure supplement 2B*). This could be due to residual binding of SET-9 and SET-26 that was statistically unable to be detected by the MACS2 pipeline (*Figure 6—figure supplement 4B*). We further examined the observed 'spreading' of H3K4me3 marking by producing meta plot of the H3K4me3 peaks that overlapped with SET-9 and SET-26 binding that centered around the summits of SET-9 and SET-26 binding peaks. We again observed an insignificant increase at the center of the plot, where SET-9 and SET-26 binding peaked, but a detectable increase of H3K4me3 levels in regions flanking the summit of SET-9 and SET-26 binding sites in the F1 and F3 *set-9 set-26* double mutant compared to wild-type worms (*Figure 8—figure supplement 3A*).

It is interesting to note that the F3 *set-9 set-26* double mutant showed a more obvious expansion of H3K4me3 compared to the F1 *set-9 set-26* double mutant, even though this difference was not statistically significant, as their 95% confidence intervals overlapped (*Figure 8C and D* and *Figure 8—figure supplement 3A*). Since the F3 *set-9 set-26* double mutant had a more severe fertility defect (*Figure 2*), this observation hinted at a possible correlation between H3K4me3 expansion and germline defects. We previously reported that the levels of H3K4me3 as detected by Western blotting was not impacted by the loss of *set-26* in the germlineless *glp-1* mutant (*Ni et al., 2012*),

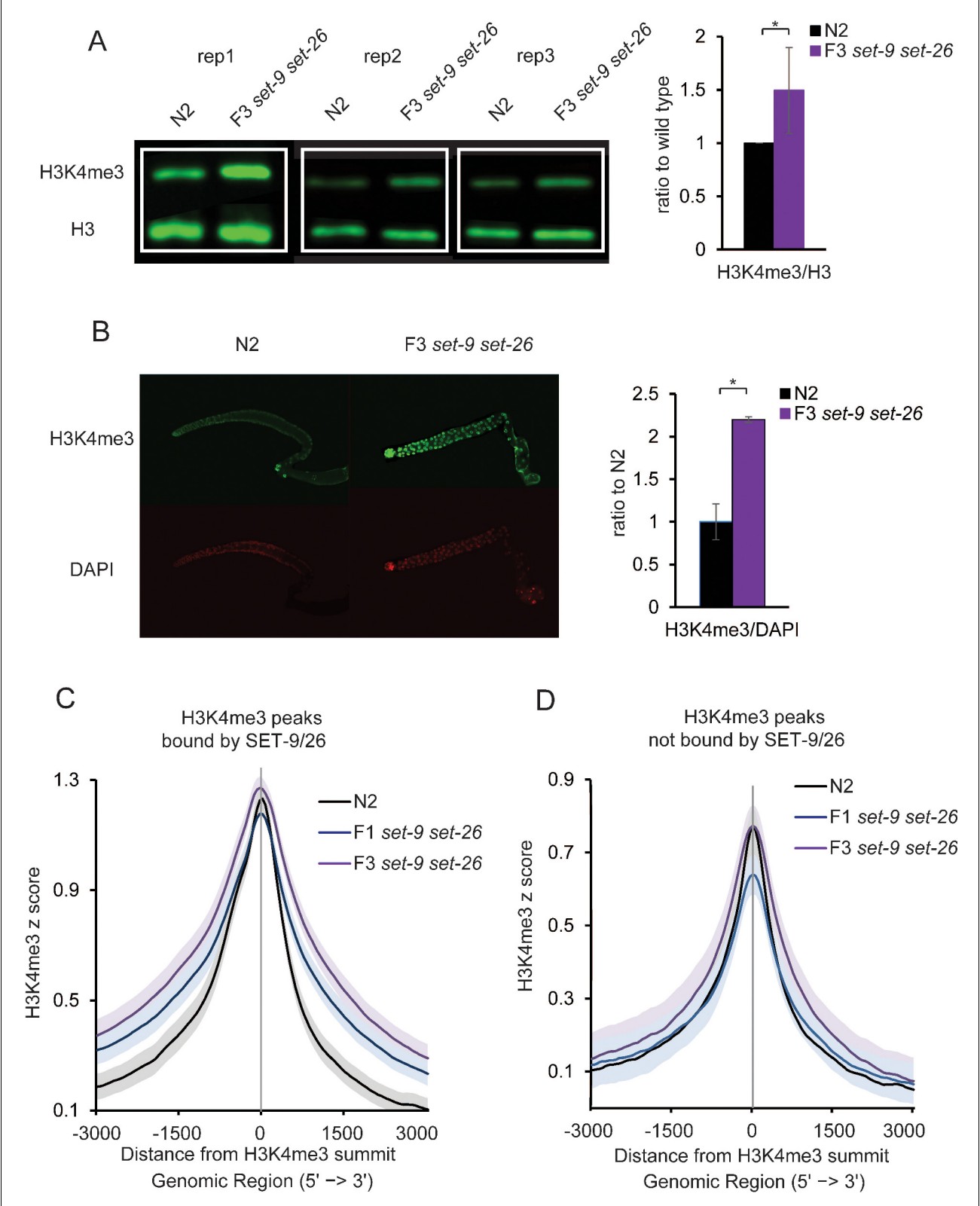

**Figure 8.** SET-9 and SET-26 restrict the spreading of H3K4me3. (**A**) Western blotting of H3K4me3 levels in three independent replicates of wild type and F3 *set-9(rw5) set-26(tm2467)* double mutant (left). Synchronized L4 worms for N2 and the F3 *set-9(rw5) set-26(tm2467)* double mutant were used. Quantification of normalized H3K4me3 levels from three independent Western experiments (right). (*p<0.05) (**B**) Elevated H3K4me3 levels were observed in the dissected gonads of the F3 *set-9(rw5) set-26(tm2467)* double mutant. Synchronized D1-D2 adults for N2 and the F3 *set-9(rw5) set-26*

*Figure 8 continued on next page*

*Figure 8 continued*

(tm2467) double mutant were used. Representative immunostaining images are shown. Right panel shows quantification of images shown on left. (*p<0.05) (C) Metagene plots showing the average H3K4me3 z-score (standardized log2 ratios of ChIP/H3ChIP signals) for the peaks bound by SET-9 and SET-26 in N2 (black) and the F1 (blue) and F3 (purple) set-9(rw5) set-26(tm2467) double mutants. Regions 3000 bp upstream and downstream of the peak summits are shown. The grey areas indicate 95% confidence intervals. (D) Metagene plots showing the average H3K4me3 z-score for the peaks not bound by SET-9 and SET-26 in N2 (black) and the F1 (blue) and F3 (purple) set-9(rw5) set-26(tm2467) double mutant. The grey areas indicate 95% confidence intervals.

DOI: https://doi.org/10.7554/eLife.34970.021

The following figure supplements are available for figure 8:

**Figure supplement 1.** SET-9 and SET-26 restrict the spreading of H3K4me3.

DOI: https://doi.org/10.7554/eLife.34970.022

**Figure supplement 2.** SET-9 and SET-26 restrict the spreading of H3K4me3 (part 2).

DOI: https://doi.org/10.7554/eLife.34970.023

**Figure supplement 3.** SET-9 and SET-26 restrict the spreading of H3K4me3 (part 3).

DOI: https://doi.org/10.7554/eLife.34970.024

**Figure supplement 4.** Germline-specific genes show correlated changes in H3K4me3 and RNA expression.

DOI: https://doi.org/10.7554/eLife.34970.025

suggesting that inactivation of SET-26 does not impact the global levels of H3K4me3 in the soma. To further investigate this possibility, we performed similar H3K4me3 ChIP-seq analysis in the germlineless glp-1; set-26 and glp-1 mutants. Interestingly, we observed no significant difference in the genome-wide patterns of H3K4me3 in germlineless worms with or without SET-26, even in regions bound by SET-9 and SET-26 (*Figure 8—figure supplement 3B*). The ChIP-seq results therefore corroborated with our previous Western blotting results, and together they supported the notion that SET-9 and SET-26 binding does not detectably impact H3K4me3 marking in somatic cells. Rather, SET-9 and SET-26 binding likely normally help to restrict H3K4me3 marking specifically in the germline.

## A subset of germline-specific genes show correlated changes in H3K4me3 and RNA expression levels in the F1 and F3 set-9 set-26 double mutants

H3K4me3 has been generally associated with active gene expression (*Sims et al., 2007*). We wondered whether the regions with expanded H3K4me3 marking could be associated with gene expression changes in the F1 and F3 set-9 set-26 double mutants. To assess this, we used csaw, an R package for differential binding analysis of ChIP-seq data using sliding windows, to identify 3438 and 5456 regions that exhibited statistically significant different H3K4me3 marking in F1 and F3 set-9 set-26 double mutants compared with wild-type worms (data not shown). Consistent with a slight global increase in H3K4me3 levels in the F1 and F3 set-9 set-26 double mutant, 92 and 99% of the differential H3K4me3 regions revealed by csaw showed elevated H3K4me3 levels in the F1 and F3 set-9 set-26 double mutants. We next assigned these differential H3K4me3 peaks to their closest genes and identified 3017 and 4517genes that are associated with altered H3K4me3 markings. We then compared these gene lists with the list of genes that exhibited significant expression change between the F1 and F3 set-9 set-26 double mutants compared to wild-type worms based on our earlier RNA-seq data. In this comparison, only 479 and 310 genes with elevated H3K4me3 markings showed expression change in the F1 and F3 set-9 set-26 double mutants (*Figure 8—figure supplement 3C and D*), an overlap that was significantly lower than what would be expected based on random chance. These results suggested that the expanded H3K4me3 regions were not generally accompanied with detectable gene expression changes based on comparison with RNA-seq analyses from whole worms.

Since our data suggested that the 'spreading' of H3K4me3 surrounding SET-9 and SET-26 binding sites likely occur specifically in the germline, we wondered whether the SET-9 and SET-26 target genes in the germline would show a correlation between H3K4me3 'spreading' and gene expression increase. To investigate this possibility, we compared the H3K4me3 profiles for the genes that were bound by SET-9 and SET-26 and showed expression increase in F1 set-9 set-26 double mutant compared with wild-type worms (*Figure 7*), and were determined to be 'germline-specific' based on

previous reports (*Reinke et al., 2004*) (*Figure 5* and *Figure 8—figure supplement 4A*). We found that their average H3K4me3 markings were highly upregulated in the F1 and F3 *set-9 set-26* double mutants compared with wild-type worms, and the elevation was noticeable at both the TSS and the TES and throughout the gene body (*Figure 8—figure supplement 4B*). A comparison of this list of germline-specific SET-9 and SET-26 target genes also indicated that their H3K4me3 elevation (based on csaw) was significantly correlated with gene expression increase (based on RNA-seq) (*Figure 8—figure supplement 4C*). Therefore, it appeared that for the germline-specific genes, H3K4me3 expansion indeed correlates well with RNA expression increase. We concluded that SET-9 and SET-26 have a unique function in the germline, where they likely restrict H3K4me3 domains and maintain the expression of a subset of genes.

### RNAi screens revealed that SET-2, the H3K4me3 methyltransferase, functions cooperatively with SET-9 and SET-26 to regulate germline function

Since many histone methyltransferases and demethylases are known to play important roles in germline function in *C. elegans* (*Katz et al., 2009*; *Li and Kelly, 2011*; *Nottke et al., 2011*; *Xiao et al., 2011*; *Robert et al., 2014*; *Kerr et al., 2014*; *Greer et al., 2014*), we performed an RNAi screen targeting putative histone methyltransferases and demethylases to uncover genes that potentially work with SET-9 and SET-26 to regulate germline function. We treated F1 *set-9 set-26* double mutant worms, which showed a mild reproductive defect (*Figure 2*), with each of the RNAi and assayed their consequent brood size. Whereas most of the RNAi treatment did not substantially affect the brood size of the F1 *set-9 set-26* double mutant, we found that three components of the MLL complex, which is well-established to deposit H3K4me3, including *set-2*, *set-16* and *wdr-5.1*, significantly reduced the brood size of the F1 *set-9 set-26* double mutant when knocked down (*Figure 9A*). To rule out off-target effects, we crossed the partial loss-of-function *set-2(ok952)* mutant with the *set-9 set-26* double mutant and assayed their brood size at the F1 generation. We again observed a synergistic effect, where the *set-2(ok952)* single mutant had a normal brood size as previously reported, but the F1 *set-2; set-9 set-26* triple mutant had a drastically reduced brood size compared with the F1 *set-9 set-26* double mutant (*Figure 9B*). This result is consistent with the model that SET-2 and SET-9 and SET-26 regulate H3K4me3 marking in different ways and their simultaneous loss results in synergistic detrimental effect. Our data above indicated that SET-9 and SET-26 bind to H3K4me3 and restrict H3K4me3 domain, likely specifically in the germline. The results with the weak loss-of-function *set-2* mutation suggested that suboptimal deposition of H3K4me3 together with broadening of H3K4me3 marking could cause substantial defects in germline function.

## Discussion

In this study, we revealed that the highly homologous paralogs SET-9 and SET-26 have unique as well as redundant functions. We found that SET-9 and SET-26 share redundant function in germline development, but only SET-26, not SET-9, plays a role in modulating lifespan and resistance to heat stress. We confirmed that SET-9 is only detectable in the germline whereas SET-26 is broadly expressed, and we concluded that their differential expression patterns likely account for the different phenotypes associated with the loss of each gene. Our transcriptomic analyses corroborated previous genetic studies and implicated SET-26 to act through DAF-16 in somatic cells to modulate longevity. We further demonstrated that SET-9 and SET-26 bind to H3K4me3 in vitro and in vivo, and that the loss of *set-9* and *set-26* results in the broadening of H3K4me3 domains surrounding most SET-9 and SET-26 binding regions, likely specifically in the germline. The expanded H3K4me3 domains were associated with increased expression of germline-specific genes bound by SET-9 and SET-26. We propose that in the soma, SET-9 and SET-26 somehow impact DAF-16 activities to modulate longevity, whereas in the germline, they are critical for restricting H3K4me3 domains and regulating the expression of specific target genes with important consequence on germline development (*Figure 10*).

### *set-9* and *set-26* are duplicated genes

*set-9* and *set-26* share high sequence identity both within the coding regions and in the non-coding sequences flanking the coding regions (near 90% identity in the ±500 bp regions), and even in the

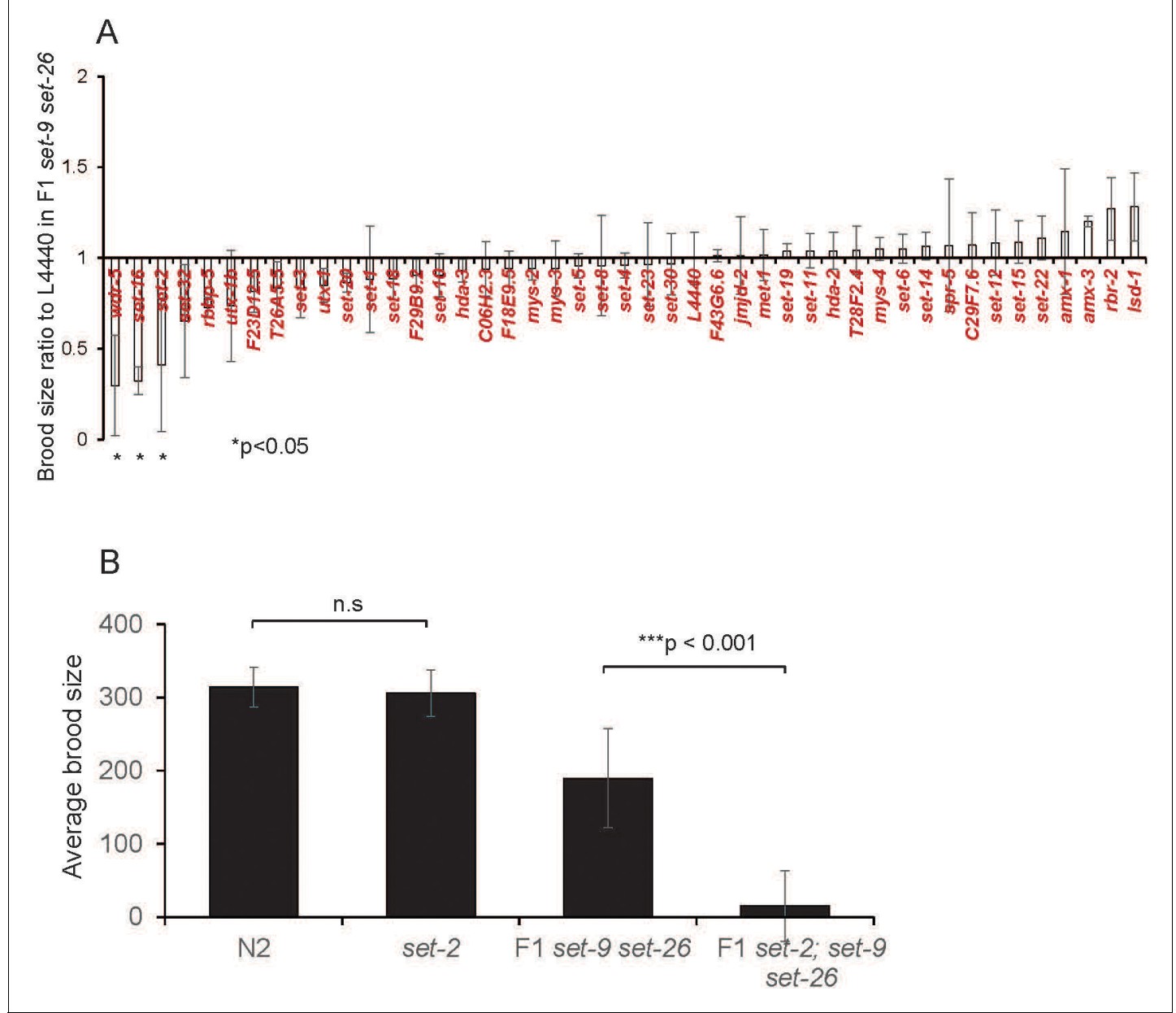

**Figure 9.** SET-9 and SET-26 and SET-2 act synergistically to regulate fertility. (**A**) Ratio of eggs laid by F1 *set-9(rw5) set-26(tm2467)* double mutant worms fed dsRNA of *C. elegans* potential histone modifiers (mainly methyltransferases and demethylases) or empty vector (L4440) for one generation. (**B**) Average brood size in N2, *set-2(ok952)*, F1 *set-9(rw5) set-26(tm2467)* double mutant and F1 *set-2(ok952); set-9(rw5) set-26(tm2467)* triple mutant.
DOI: https://doi.org/10.7554/eLife.34970.026

genes 3' of *set-9* and *set-26* (Y24D9B.1 and Y51H4A.13 respectively). Although *set-9 and set-26* are highly conserved from worms to yeast to mammals, only one homolog in each of the other diverse species has been detected based on sequence and domain structure alignment (*Rincon-Arano et al., 2012*; *Zhang et al., 2017*). Even the closely related *Caenorhabditis* species, such as *C. briggsae*, *C. remanei*, *C. brenneri* and *C. japonica*, only harbor one gene that is highly homologous to *set-9 and set-26*. These results suggested that *set-9* and *set-26* have arisen from gene duplication specifically in *C. elegans*. In the future, it will be interesting to explore how this duplication event allowed *set-9* and *set-26* to adopt unique functions.

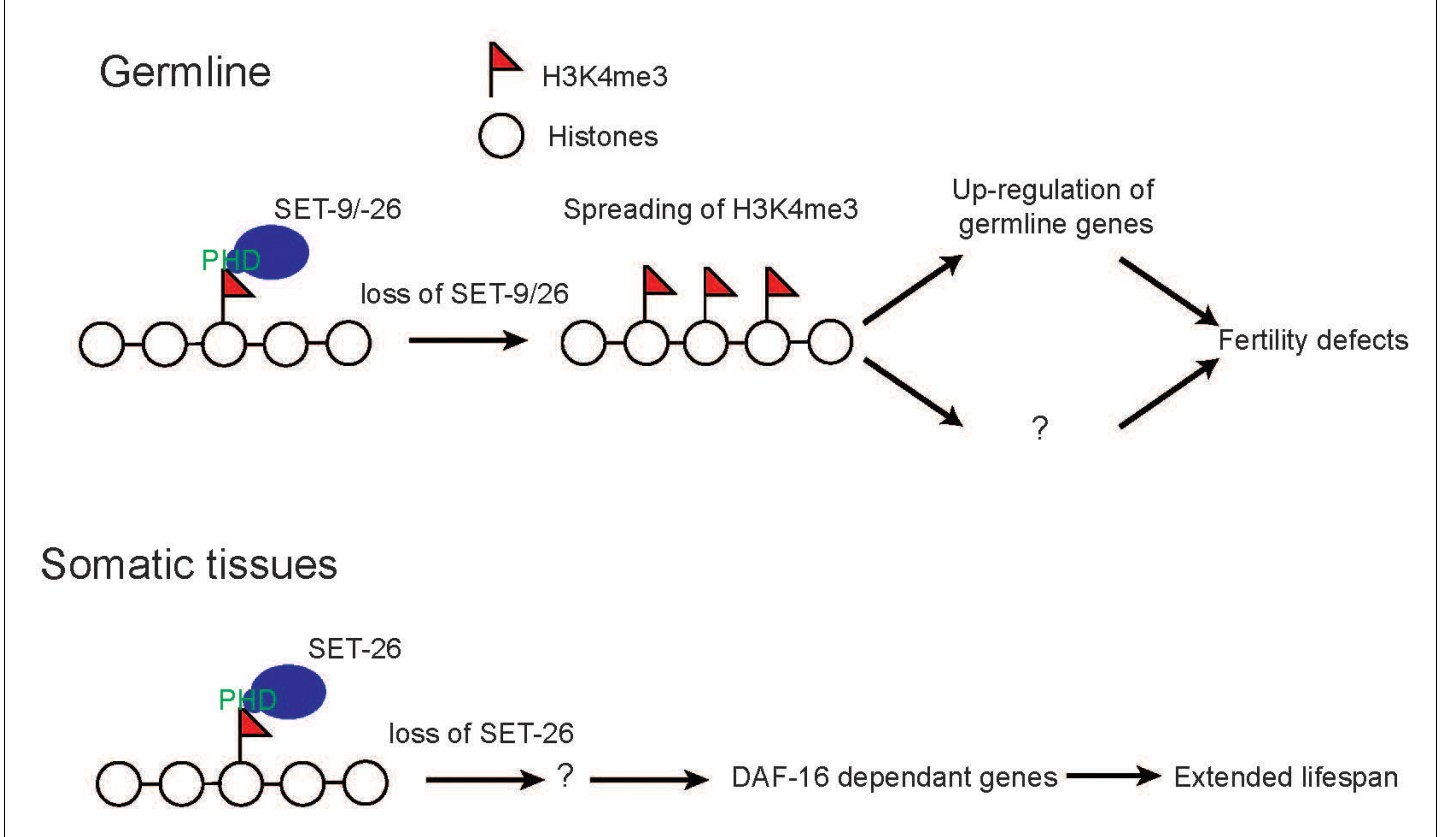

**Figure 10.** SET-9 and SET-26 regulate H3K4me3 and target gene expression. In the germline, loss of *set-9* and *set-26* results in the broadening of H3K4me3 domains surrounding most SET-9 and SET-26 binding regions and up-regulation of germline genes. In the soma, loss of *set-26* modulates lifespan by indirectly regulating DAF-16-dependent genes. We propose that SET-9 and SET-26 are critical for organizing local chromatin environment and regulating the expression of specific target genes, and these activities together contribute to their roles in germline development and longevity.
DOI: https://doi.org/10.7554/eLife.34970.027

## SET-9 and SET-26 in chromatin regulation

Based on the similarity in sequence and domain structure, *Drosophila* UpSET and mammalian SETD5 and MLL5 represent the likely homologs of SET-9 and SET-26. SET-9 and SET-26, UpSET, SETD5 and MLL5 all harbor centrally localized SET and PHD domains, and they share ~30–40% sequence identity in their PHD and SET domains.

The SET domains of SET-9, SET-26, UpSET, SETD5 and MLL5 proteins are highly conserved and all share similar mutations that suggest they should lack enzymatic activities (*Figure 6—figure supplement 1A*). Indeed, UpSET, SETD5 and MLL5 have not been found to harbor methylating activities (*Osipovich et al., 2016*; *Rincon-Arano et al., 2012*; *Madan et al., 2009*). Whereas the SET domain of SET-26 has been suggested to methylate H3K9me3 in vitro (*Greer et al., 2014*), our data indicated that SET-9 and SET-26 do not have major roles in H3K9me3 deposition in *C. elegans*. The exact functions of the SET domains in these proteins remain to be elucidated.

The PHD domains of SET-9, SET-26, UpSET and MLL5 proteins are also highly conserved. The PHD domain of MLL5 was found to bind to H3K4me3 in vitro (*Ali et al., 2013*), and both MLL5 and UpSET have been shown to localize at promoter regions in cultured cells (*Rincon-Arano et al., 2012*; *Ali et al., 2013*). Our results demonstrated that SET-9 and SET-26 also bind to H3K4me3 in vitro and in vivo. We further revealed that the PHD domains of SET-9 and SET-26 bind to H3K4me3 with adjacent acetylation (e.g. K9, K14, K18) with much greater affinity (*Figure 6B*). This is an important finding and potentially links SET-9 and SET-26 to the regulation of both H3K4me3 and histone acetylation. Indeed, we showed that loss of *set-9* and *set-26* results in a slight elevation of global H3K4me3 and H3K9Ac levels (*Figure 8A* and *Figure 8—figure supplement 1B*). We additionally

uncovered expansion of H3K4me3 domains surrounding SET-9 and SET-26 bound regions, and we predict that similar expansion of H3K9Ac likely occurs in the *set-9 set-26* double mutant. Interestingly, loss of *set-26* in somatic tissues does not result in similar H3K4me3 elevation (*Figure 8—figure supplement 1F*) and we concluded that SET-9 and SET-26 are particularly important for restricting H3K4me3 domains in the germline. These data are reminiscent to those reported for MLL5 and UpSET (*Rincon-Arano et al., 2012*; *Gallo et al., 2015*). In human glioblastoma cells with self-renewing potential, it was found that knockdown of MLL5 leads to increased global levels of H3K4me3 and a more open chromatin environment (*Gallo et al., 2015*). Importantly, this anti-correlation between MLL5 and H3K4me3 was only detected in primary glioblastoma cells with self-renewing potential, but not in bulk glioblastoma samples, nor in non-neoplastic brain samples or colon cancer cells (*Gallo et al., 2015*), indicating a regulatory process that is highly cell type specific. A possible parallel in *C. elegans* is that the anti-correlation between SET-9 and SET-26 status and H3K4me3 levels is particularly obvious in the germline of adult worms (*Figure 8B*), which harbor proliferative germline stem cells. In *Drosophila* Kc cells, UpSET knockdown has been shown to increase the levels and spreading of H3K9Ac, H3K16Ac, and H3K4me2/3 around some TSSs, which results in a generally more accessible chromatin environment (*Rincon-Arano et al., 2012*). UpSET achieves this partly through recruitment of specific histone deacetylases. Although the role of SETD5 PHD domain has not been studied, SETD5 also has been found to interact with histone deacetylase complex (*Osipovich et al., 2016*). Loss of SETD5 causes elevation in histone acetylation at transcriptional start sites and near downstream regions (*Osipovich et al., 2016*), suggesting that it acts in a similar manner to *Drosophila* UpSET. Taking together the *C. elegans*, *Drosophila*, and mammalian data, it appears that the SET-9, SET-26, UpSET, SETD5, MLL5 family of factors play an important role in binding to H3K4me3, possibly with flanking acetylation sites, and regulating local chromatin accessibility, partly through restricting the spread of histone modifications such as H3K4 methylation and H3K9 acetylation. Extending from the findings with UpSET in *Drosophila* and SETD5 in mammals, SET-9 and SET-26 likely also recruit demethylating and deacetylating enzymes to confine local methylation and acetylation domains.

## Biological functions of SET-9 and SET-26

In this study, we uncovered a novel redundant function of SET-9 and SET-26 in germline function. It is interesting to note that an earlier report suggested that loss of *set-26*, but not *set-9*, accelerated the progressive sterility of the *spr-5* mutant (*Greer et al., 2014*). However, our analyses indicated that the *set-9(n4949)* mutant used in the study (*Greer et al., 2014*) represents a deletion/duplication allele, as we can PCR amplify the *set-9* gene from the mutant (data not shown). Our data using the newly generated *set-9(rw5)* deletion/frame-shift mutant supported a role of SET-9 in collaboration with SET-26 to regulate germline development.

In considering the possible homologs of SET-9 and SET-26 in other species, MLL5 has been implicated in male fertility (*Heuser et al., 2009*; *Madan et al., 2009*), whereas UpSET is important for female fertility (*Rincon-Arano et al., 2012*). Our data indicated that SET-9 and SET-26 have a more pleiotropic role in the germline, affecting both germline stem cell proliferation, and the subsequent differentiation into oocytes and sperms. This difference could be due to that *C. elegans* are hermaphrodites and SET-9 and SET-26 have adopted broader functions in the germline. In addition, MLL5 has been implicated in regulating cell cycle progression (*Deng et al., 2004*) and stem cell pluripotency (*Zhang et al., 2009*), which may parallel the redundant roles of SET-9 and SET-26 in germline stem cells proliferation. Considering the highly conserved functions of SET-9, SET-26, UpSET, SETD5 and MLL5 at the molecular and phenotypic levels, an intriguing possibility is that UpSET, SETD5 and MLL5 have yet-to-be characterized roles in stress response and longevity.

A key important question is how SET-9 and SET-26 can mediate their effects on stress response, longevity and germline development. Our data clearly supported that SET-26 acts in the soma and through DAF-16 to modulate longevity. The exact functional relationship between SET-26 and DAF-16 remains unclear. It was striking that the comparison of the SET-9 and SET-26 bound targets from whole worms and the DAF-16-mediated transcriptomic changes in germlineless mutant not only did not show a significant overlap, it in fact showed an overlap that was much lower than expected by random chance. Although it is difficult to cross-compare results from different experimental set-ups, it seems likely that SET-9 and SET-26 do not bind to candidate DAF-16 regulated genes. Therefore, SET-9 and SET-26 likely mediate their effect on DAF-16 indirectly.

We also note that while we were able to identify SET-9 and SET-26 targets whose expression show significant change in response to the loss of *set-9* and *set-26*, the SET-9 and SET-26 binding targets were not enriched for genes that showed expression change in the *set-9 set-26* double mutant. In other words, genes bound by SET-9 and SET-26 were not more likely to show expression change when *set-9* and *set-26* were deleted. Since our ChIP-seq data were from whole worms and could not distinguish the SET-9 and SET-26 binding targets in the soma vs the germline, our current results cannot resolve whether SET-9- and SET-26-binding has a direct consequence on gene expression regulation.

We detected significant broadening of H3K4me3 marking surrounding the SET-9 and SET-26 bound regions. Although this change was observed in whole worms, it was not detectable in worms lacking germline (*Figure 8—figure supplement 1F*). We interpreted this to mean that the expansion of H3K4me3 domains likely happen either specifically in the germline, or most noticeable in the germline. Somewhat surprisingly, such broadening of H3K4me3 domains did not appear to correlate with gene expression using whole worm analyses. This finding could partly be due to technical caveats, as the ChIP-seq and RNA-seq data were generated using whole worms, which could mask correlated changes in specific cells. Consistent with this speculation, we detected a significant and positive correlation between H3K4me3 expansion with RNA expression change of germline-specific genes. In the future, ChIP-seq and RNA-seq analyses using dissected gonads *vs* somatic tissues will help to further establish this prediction.

In addition to gene regulation, the expanded H3K4me3 regions likely represent perturbed chromatin environment that could interfere with processes other than gene regulation, such as genome maintenance. In fact, in *C. elegans*, altered H3K4 methylations have been shown to predispose mutant worms to DNA damage and genome instabilities, which result in germline defects (*Nottke et al., 2011*). Interestingly, inactivating UpSET in *Drosophila* and MLL5 in mammals also lead to increased genome instability and DNA damage (*Tasdogan et al., 2016*; *Rincon-Arano et al., 2012*). In *C. elegans*, we observed an increased number of germ cell apoptosis in the *set-9 set-26* double mutant (data not shown). Moreover, RNAi knockdown of *mre-11*, a double-strand break repair protein, synergistically aggravated the fertility defects of the *set-9 set-26* double mutant (data not shown). These data together suggested that DNA damage and genome instability could be increased in the *set-9 set-26* double mutant, which may contribute to their germline defects.

In summary, our findings provided new mechanistic insights into the functions of SET-9 and SET-26 with important implications for their roles in longevity and germline development. We revealed that SET-9 and SET-26 are recruited to H3K4me3 marked regions and participate in confining H3K4me3 domains, particularly in the germline. Loss of *set-9* and *set-26* in these regions likely leads to a more open and accessible chromatin environment and RNA expression change in a subset of genes. In addition, SET-26 acts in the soma to modulate longevity and it achieves this by regulating DAF-16-mediated transcription indirectly. Our findings are consistent with the possibility that human MLL5 and SETD5 represent functional homologs of SET-9 and SET-26, and the results discussed here provide new insights into how MLL5 and SETD5 may act and also implicate them in stress response and longevity.

## Materials and methods

### Worm strains
The N2 strain was used as the wild-type (WT). The mutants used in the study are: *set-2(ok952)*, *set-9 (rw5)*, *set-26(tm2467)*, *rrf-1(pk1417)*, *daf-16(mgDf47)*, *glp-1(e2141)*; the *set-9(rw5) set-26(tm2467)* double mutant was maintain as balanced heterozygote *set-9(rw5) set-26(tm2467)*/nT1. The GFP knock-in strains constructed in this study are: *set-9::gfp(rw24)*, *set-26::gfp(rw25)*.

### Antibodies
The antibodies used were anti-H3K4me3(Millipore, Billerica, MA 17–614), anti-H3K9me3(abcam, Cambridge, United Kingdom ab8898), anti-GFP(abcam, Cambridge, United Kingdom ab290), anti-H3(abcam, Cambridge, United Kingdom ab1791), anti-H3K9ac(Wako, Richmond, VA 309–32379).

## Brood size assay

Brood size assay was performed as described (*Li et al., 2008*). Each single L4 worm was picked onto individual plate and was transferred to a new plate every 24 hr until the end of its reproductive phase. Dead eggs and alive progenies were counted as its total brood size. All experiments were repeated two to three times. Student's t-test was used to calculate the p-values.

## Lifespan assay

Lifespan assay was performed as described (*Li et al., 2008*). All experiments were performed at 20°C. For RNAi plates, the *set-9/26* RNAi construct was taken from the Ahringer RNAi library. RNAi bacteria were grown in LB with 100 ug/ml Carbenicillin (Carb) and 15 ug/ml Tetracycline (Tet) at 37°C to OD600 around 0.8. The culture was concentrated 5-fold, and seeded onto plates with Carb and Tet. Sufficient IPTG stock was added to plates so that final IPTG concentration is 4 mM. Let plates dry and induce for ~4 hr before use. For lifespan assays using NGM plates seeded with OP50 bacteria, the OP50 bacteria were grown in LB overnight at 37°C and the culture was concentrated 3-fold and seeded onto plates. For all lifespan assays, Worms were picked onto RNAi plates to lay ~40 eggs per plate and the progeny were grown on the plate until they reached adulthood. The worms were transferred to a new plate every day during their reproductive phase and then transferred to a new plate every 4 days. Worms were scored every other day, and those that failed to respond to a gentle prodding with a platinum wire were scored as dead. Animals that bagged, exploded, or crawled off the plate were considered as censored. We defined the day when we transferred the young adult worms as day 0 of adult lifespan. All the lifespan experiments were repeated at least two independent times. Standard survival analysis was performed using SPSS and OASIS software (*Yang et al., 2011*). The survival function was estimated using the Kaplan-Meier method, and the p-values were calculated using a log-rank test.

## Heat stress assay

Heat stress assay was performed as described (*Li et al., 2008*). Synchronous D0 adult worms grown on OP50-NGM plates at 20°C were shifted to 35°C. Worms were scored every 3–4 hr. All the heat stress experiments were repeated at least two independent times. All experiments were repeated two to three times. Standard analysis was performed using OASIS software. The survival function was estimated using the Kaplan-Meier method, and the p-values were calculated using a log-rank test.

## Analysis of germline mortality

Mortal Germline assays were performed by transferring 6 L1 larvae per plate to fresh NGM plates every generation, as previously described (*Ahmed and Hodgkin, 2000*). Percentage fertile lines were calculated as the number of fertile plates divided by the number of total plates.

## Immunofluorescence staining

The gonads from 100 to 150 worms were dissected out using syringe needle on a poly-lysin coated slide. Gonads were permeabilized by the standard freeze-crack method and were fixed in 4% formaldehyde fixative (PBS/4% formaldehyde) for 30 min followed by 5 min incubation in chilled methanol at −20°C in a Coplin jar. Slide was washed twice with PBST for 5 min each and finally incubated with blocking buffer (PBS + 5% BSA + 0.1% tween-20 +0.1% triton-100) for 1 hr. Blocking buffer was removed and gonads were incubated with 25 µl of 1:50 dilution of H3K4me3 antibodies for overnight at 4°C in a humidity chamber. Next day, gonads were washed twice with PBST for 10 min each and incubated with 25 µl of 1:50 dilution of secondary antibody for 1–2 hr at RT. DAPI (final concentration 25 ng/ µl) was also added with secondary antibody. Gonads were further washed three times with PBST for 10 min each and mounted in 10 µl vectashield mounting medium (Vecta Laboratories). Experiments were repeated twice. 4 worms and 5–7 germline nuclei for each worm were used for quantification of H3K4me3 and DAPI. Student's t-test was used to calculate the p-values.

## DAPI staining and measuring the number of germ cells within the mitotic region

The gonads from 100 to 150 worms were dissected out using syringe needle. Gonads were fixed in 4% formaldehyde fixative (PBS/4% formaldehyde) for 1 hr. After the removal of fixative, gonads were washed twice with 1 ml PBST (PBS buffer +0.1% Tween 20) each. Fixed worms were incubated with DAPI (25 ng/ μl in PBS) for 30 min. Gonads were further washed three times with 1 ml PBST for 5 min each and mounted on agarose pads in 10 μl vectashield mounting medium (Vecta Laboratories). DAPI-stained gonad images were taken using the Z-stacking function of the microscope. To count mitotic germ cell numbers, we marked the boundary between mitotic zone and transition zone by observing the transition zone-specific nuclear morphology (crescent shape) and counted the number of nuclei in each focal plane within the mitotic region. Experiments were repeated three times. 6–9 worms were used for quantification. Student's t-test was used to calculate the p-values.

GST protein pull-down assay pGEX-2TK was used for generating bacteria expression construct. SET-9/26$_{PHD}$ cDNA was amplified from wildtype cDNA by PCR using the primers: SET-9/26$_{PHD}$-F: C TCAGGATCCGATTCCGAATCCGAGGGAA; SET-9/26$_{PHD}$-R: GCGTGAATTCCCGCTCGAAGTCGA TTCAAAA; and subcloned into pGEX-2TK.

The expression construct was transformed into BL21 bacteria. Bacteria containing cDNA of SET-9/26$_{PHD}$ was culture to OD600 equal to around 0.6 and the SET-9/26$_{PHD}$ expression was induced by adding IPTG to final concentration equals to 0.5 mM. Bacteria were lysed by sonication and protein was purified by Glutathione Sepharose (Sigma).

GST protein pull-down assay was performed as described (*Tsai et al., 2010*). 25 μg GST-PHD protein were incubated with 10, 100, 1000 μg of calf thymus total histones (Sigma) in 500 μl NTP overnight at 4°C. 90 μl of a 50% slurry of GST-beads were added and incubated for 2 hr at 4°C, recovered by centrifugation and washed six times (10 min at 4°C) with NTP buffer (50 mM Tris-HCl 7.4, 300 mM NaCl, 0.1% NP-40). The protein bound beads were analyzed by SDS-PAGE and detected by Coomassie stain.

## Array binding assay

The array binding assay was performed by EpiCypher. Briefly, GST-PHD were applied on an EpiTitan array that was separated by a gasket such that two chambers were delineated. After the protein incubation, a series of the anti-GST (primary) and the fluorescent AlexaFluor 647 (secondary) antibody incubation steps were carried out to detect the bound protein. Two independent experiments were performed.

## Immunoblotting and quantification

Immunoblotting was performed as described (*Ni et al., 2012*). Synchronized embryos were put onto NGM-OP50 plates and grown at 20°C until reaching mid-L4. Worms were harvested and washed three times using ice-cold M9 buffer. Worm pellets were lysed with boiling SDS sample and equal amount of lysates were used for SDS-PAGE(18%) and transferred to nitrocellulose membranes. The membranes were incubated with primary antibodies overnight (H3K4me3/H3K9me3/H3K9ac, 1:1,000; H3, 1:2,000). IRDye secondary antibody was used and the result was quantified using Odyssey imaging system. All experiments were repeated 3–4 times. Modified histone levels were normalized to H3 levels. Student's t-test was used to calculate the p-values.

Plasmid and homologous DNA repair template construction pU6::set-9 sgRNA was generated using pU6::unc-119 sgRNA as described (*Friedland et al., 2013*). The pU6::unc-119 sgRNA was used as template to amplify two overlapping PCR fragments using the primers U6prom EcoRI F and set-9 gRNA R or set-9 gRNA F and U6prom HindIII R. These PCR products were gel-purified and then mixed together in a second PCR with primers U6prom EcoRI F and U6prom HindIII R. This final PCR product was digested with EcoRI and HindIII and ligated into a pU6::unc-119 sgRNA plasmid that had been digested with EcoRI and HindIII, creating the vector pU6::set-9 sgRNA. The pU6::set-9/set26 sgRNAs (1 and 2) that used for generating *set-9::gfp* and *set-26::gfp* strains were constructed using the same strategy. The homologous DNA repair templates that used for generating *set-9::gfp* and *set-26::gfp* strains were designed and synthesized as described (*Paix et al., 2014*). Homologous arms flanking gfp DNA sequence were generated using the pPD90 plasmid that contains gfp sequence as template and the primers: armF12-GFP: CGAGACGAAGCCGaTCtACgCG

aTGGAAcagtaaaggagaagaacttttcactggagttg; armR12-GFP: caagttttttcgcagattccttgCTAtttgtatagttcA tccatgccatgtgtaatccc; The DNA repair template with ~30 bp homologous arms was generated using the primer: armF1-complete: ccctcaatttttttcagCTGAAACAAACTCGAGACGAAGCCG; and armR1-complete: gggacaatttttattcttcaagttttttcgcagattcc. The DNA repair template with ~60 bp homologous arms was generated using the primer: armF2-complete: ccaaaaaatctccttaaaaaccctcaatttttttcagC TGAAACAAACTCGAGACGAAGC;                    and                    armR2-complete: cgagatagaaagagatgatatgggacaatttttattcttcaagttttttcgcagattcc.

## CRISPR-mediated genome editing

CRISPR-mediated genome editing was performed as described (*Paix et al., 2014*; *Friedland et al., 2013*). For generating *set-9* mutant, day1 adult animals were injected with pDD162 (Peft-3::Cas9:: tbb-2 3'UTR), pCFJ90 (pmyo-2::mCherry) and pU6::set-9 sgRNA and grown overnight at 16°C. The survived worms were separated, transferred to 20°C and their F2 mCherry-positive animals were genotyped for *set-9* mutation. For generating *set-9::gfp* and *set-26::gfp* strains, day1 adult animals were injected with pDD162 (Peft-3::Cas9::tbb-2 3'UTR), pCFJ90 (pmyo-2::mCherry), pU6::set-9/ set26 sgRNAs (1 and 2) and homologous DNA repair templates and grown overnight at 16°C. The survived worms were separated, transferred to 20°C and their F2 mCherry-positive animals were genotyped for GFP knock-in strains. sgRNAs that target two loci and two DNA repair templates with different length of homologous arms (one ~30 bp and the other ~60 bp) were co-injected to increase efficiency. The *set-9::gfp* and *set-26::gfp* strains were mounted on a microscope slide and visualized using a Zeiss 710 confocal system.

## Chromatin immunoprecipation following sequencing (ChIP-seq)

ChIP experiments for SET-9::GFP and SET-26::GFP were performed as described (*Zhong et al., 2010*) with the following modifications. Approximately 70,000–100,000 L4s were harvested and crosslinked in 2% formaldehyde-M9 solution for 25 min at room temperature with rotation. The worms were then washed with 100 mM Tris pH 7.5 to quench formaldehyde solution, washed two times with M9, and once with FA buffer (50 mM HEPES/KOH pH 7.5, 1 mM EDTA, 1% Triton X-100, 0.1% sodium deoxycholate; 150 mM NaCl) supplemented with 2X protease inhibitors (Roche Cat#11697498001, cOmplete Protease Inhibitor Cocktail Tablets). Worms were then collected in a 15 ml conical tube, snap-frozen in liquid N2 and stored at −80°C. The pellet was resuspended in 1 ml FA buffer plus protease inhibitors. Using a Bioruptor sonicator, the sample was sonicated on ice/ salt water 30 times with the following settings: 30 s on, 60 s off. The chromatin was then further sheared by the covaris s2 40 times with the following settings: 20% duty factor, intensity 8, 200 cycles per burst, 60 s on, 45 s off. The tube spun containing worm extract was then spun at 13,000 g for 30 min at 4°C. The supernatant was then transferred to a new tube and the protein concentration of the supernatant was then determined by Bradford assay. Extract containing approximately 2 mg of protein was incubated in a microfuge tube with 6–12 ul anti-GFP antibodies overnight at 4°C with gentle rotation. 10% of the material was removed and used as input DNA. Then 30 ul of protein A conjugated to sepharose beads (EMD Millipore) were added to each ChIP sample and rotated at 4°C for 4 hr. The beads were then spun at 2000 rpm for 1 min to collect and washed twice for five mins each at 4°C in 1 ml of FA buffer, once in FA with 500 mM NaCl and once in FA with 1M NaCl with gentle rotation. The beads were then washed in TEL buffer (0.25 M LiCl, 1% NP-40, 1% sodium deoxycholate, 1 mM EDTA, 10 mM Tris-HCl, pH 8.0) for 5 min and twice in TE for 5 min. To elute the immunocomplexes, 50 ul Elution Buffer (1% SDS in TE with 250 mM NaCl) was added and the tube incubated at 65°C for 15 min, with brief vortexing every 5 min. The beads were spun down at 2000 rpm for 1 min and the supernatant transferred to a new tube. The elution was repeated and supernatants combined. To each sample, RNAseA was added and incubated at 37°C for 15 min, and proteinase K was added and incubated for 1 hr at 55°C, then 65°C overnight to reverse crosslinks. The DNA was then purified with the Qiaquick PCR purification kit (Qiagen), and eluted with 40 ul H2O. The immunoprecipitated DNA was either checked by qPCR or subjected to high-throughput sequencing library preparation. The protocol for library preparation for SET-9/26 ChIP–Seq is NEB-Next Ultra II DNA Library Prep Kit for Illumina (NEB).

ChIP experiments for H3K4me3 and H3K9me3 were performed as described (*Pu et al., 2015*). For worm collection, 1000–2000 F1 *set-9 set-26* double mutant worms were collected from

progenies of balanced heterozygote *set-9(rw5) set-26(tm2467)*/nT1. F3 *set-9 set-26* double mutant worms were collected from progenies of F2 *set-9 set-26* double mutant worms. L4 worm pellets were ground with a mortar and pestle and cross-linked with 1% formaldehyde in PBS for 10 min at room temperature. Worm fragments were collected by spinning at 3000 g for 5 min and resuspended in FA buffer followed by sonication with Bioruptor. Chromatin extract was incubated with antibody overnight at 4°C. Precipitated DNA (10–15 ng) from each sample was used for Illumina sequencing library preparation. DNA from ChIP was first end-repaired to generate a blunt end followed by adding single adenine base for adaptor ligation. The ligation product with adaptor was size-selected and amplified by PCR with primers targeting the adaptor. Up to 12 samples were multiplexed in one lane for single-end 50-nt Illumina HiSeq sequencing.

All ChIP-seq experiments were repeated at least two times.

## ChIP-seq data analysis

The data analysis pipeline was performed as described (*Pu et al., 2015*). Low quality reads were removed using the FASTX Toolkit. The sequencing reads from two independent experiments were combined and then aligned to the WS250 *C. elegans* genome using bowtie2. PCR duplicates were removed and bam files were generated using SAMtools. The bam files were then used for calling peaks by MACS2 (combined broad and narrow peaks). A GLM model was then applied to compute the counts of all the peaks to identify significant ones using two independent replicates. H3 ChIP data were used as a control for H3K4me3 ChIP and genomic DNA input was used as a control for SET-9 and SET-26 ChIP. ChIP signals (z-score) were normalized and calculated using bamCoverage software.

For analyses using genes, each peak was associated with its closest gene. Overlapping peaks were determined by 1 bp overlap between two peaks using bedtools. For oriented meta plots, 3000 bp upstream and downstream of the summit were included and 50 bp windows were used for normalized counts in each window. The summits determined by MACS2 were assigned to their closest genes. If a gene is on the '−' strain, the normalized counts for −3000 to 3000 bp of that summit were counted in a reverse direction. For example, if a peak, which was assigned to a gene in the '−' strain, has a summit of X, the extended region for this summit in 5' to 3' direction is X + 3000 bp to X − 3000 bp; if a peak, which has assigned to a gene in the '+' strain, has a summit of Y, the extended region for this summit in 5' to 3' direction is Y-3000bp to Y + 3000 bp. The 95% confidence interval was calculated using bootstrap method. Briefly, 10% of the total summits were randomly selected and the mean was calculated. This random selection was repeated1000 times and the 95% confidence interval was calculated based on the estimate that these 1000 mean number follow normal distribution (*Hesterberg et al., 2005*). Fishers' exact test was used when comparing two lists of genes or peaks.

H3K9ac ChIP-seq data was downloaded from: http://data.modencode.org/cgi-bin/findFiles.cgi?download=3578

## RNA isolation and library preparation for RNA-seq

RNA isolation was performed as described (*Li et al., 2008*). For worm collection, 100–200 F1 *set-9 set-26* double mutant worms were collected from progenies of balanced heterozygote *set-9(rw5) set-26(tm2467)*/nT1. F3 *set-9 set-26* double mutant worms were collected from progenies of F2 *set-9 set-26* double mutant worms. Synchronized mid/late L4 staged worms that grown at 20°C were homogenized in 1 ml Tri-reagent for 30 min at room temperature. 0.1 mL of BCP was added to the sample and mixed well. The sample was then spun at 12,000 g 15 min at 4°C and the aqueous phase was transferred to a new tube. 0.5 ml Isopropanol was added to the sample, incubated at room temperature for 10 min and spun at 12,000 g 10 min. The RNA pellet was washed twice with 75% EtOH and dissolved in water. The RNA sample was then was then purified to remove DNA using RNeasy Mini Kit (Qiagen). The protocol for library preparation was using Ovation Human FFPE RNA-Seq Library Systems (NuGEN).

## RNA-seq data analysis

The data analysis pipeline was performed as described (*Pu et al., 2015*). Low quality reads were removed using the FASTX Toolkit. Illumina primers (adaptors) were then removed using cutadapt.

And tRNA and rRNA reads were removed using Bowtie and the remaining reads were aligned to WS250 *C. elegans* genome using TopHat2 with no novel junctions. Mapped reads were then input into Cufflinks to calculate raw counts for each gene, which were then used for differential expression analysis by edgeR. Genes with less than 20 reads mapped to them in all samples were removed and the remaining genes were used as input to test for differential expression. PCA analysis was performed using the built-in function in edgeR. 5% false discovery rate (FDR) was used to determine differential expression.

## Heatmap

Heatmaps were generated using CISTROME (*Liu et al., 2011*). H3K4me3 and H3K9me3 ChIP-seq results from wild-type and the *set-9 set-26* double mutant were used to generate the maximum signal ranked heatmaps.

## RNAi screen

RNA constructs were obtained from the Ahringher libarary. HT115 bacteria containing RNAi constructs were grown at 37°C and seeded on nematode growth medium (NGM) plates containing carbenicillin and tetracycline and dry overnight. dsRNA expression was induced by adding IPTG to a final concentration 0.4 mM. Heterozygous adult *set-9 set-26*/+worms were put on plates for 1~2 hr to lay eggs and F1 homozygous *set-9 set-26* worm was picked onto new plates with RNAi bacteria. Brood size of 3–4 RNAi treated worms were scored.

## Gene ontology analysis

Gene ontology (GO) analysis was carried out using the DAVID 6.8 Bioinformatics Database (http://david.abcc.ncifcrf.gov) (*Huang et al., 2009*).

# Acknowledgements

Some strains were provided by the CGC, which is funded by NIH Office of Research Infrastructure Programs (P40 OD010440). +/nT1[qIs51] was a gift from Dr. Jun Liu's lab (Cornell University). We thank Lin Wang (Cornell University) for help with generating *set-9(rw5)* strain, members of the Lee laboratory (Cornell University) and Charles Danko (Cornell University) for insightful discussions and manuscript reading. This work was supported by R01 grant AG024425 from the NIA to SSL.

# Additional information

## Funding

| Funder | Grant reference number | Author |
|---|---|---|
| National Institutes of Health | R01 grant AG024425 | Siu Sylvia Lee |

The funders had no role in study design, data collection and interpretation, or the decision to submit the work for publication.

## Author contributions

Wenke Wang, Conceptualization, Resources, Data curation, Formal analysis, Validation, Investigation, Visualization, Methodology, Writing—original draft, Writing—review and editing; Amaresh Chaturbedi, Validation, Visualization, Methodology; Minghui Wang, Investigation; Serim An, Data curation, Validation, Visualization; Satheeja Santhi Velayudhan, Validation; Siu Sylvia Lee, Conceptualization, Resources, Supervision, Funding acquisition, Investigation, Project administration, Writing—review and editing

## Author ORCIDs

Siu Sylvia Lee http://orcid.org/0000-0001-5225-4203

Decision letter and Author response
Decision letter https://doi.org/10.7554/eLife.34970.038
Author response https://doi.org/10.7554/eLife.34970.039

## Additional files

### Supplementary files

• Supplementary file 1. Quantitative data for experiments reported. Table S1 (A) Lifespan data for *Figure 1B* were shown. (B) Heat stress resistance data for *Figure 1C* were shown. Table S2 (A) Brood size data for *Figure 2A* were shown. (B) Mortal germline assay data for *Figure 2B* were show. (C) Maternal and paternal effect on brood size for *Figure 2C* were shown. (D) Mitotic germ cell number for *Figure 2D* were shown. Table S3 (A) Wild type lifespan data with RNAi treatment for *Figure 3B* were shown. (B) *rrf-1(pk1417)* mutant lifespan data with RNAi treatment for *Figure 3C* were shown. Table S4 (A) Lifespan data for *Figure 3—figure supplement 1A* were shown. (B) Heat stress resistance data for *Figure 3—figure supplement 1B* were shown. (C) Brood size data for *Figure 3—figure supplement 1C* were shown. Table S5 (A) Histone peptide array binding assay for *Figure 6B* were shown. Table S6 (A) qPCR RNA-seq validation for *set-26*(tm2467) compare with wild type data were shown. (B) qPCR RNA-seq validation for F3 *set-9(rw5) set-26*(tm2467) compare with wild type data were shown.
DOI: https://doi.org/10.7554/eLife.34970.028

• Supplementary file 2. Gene lists for comparison indicated gene list comparisons were shown.
DOI: https://doi.org/10.7554/eLife.34970.029

• Transparent reporting form
DOI: https://doi.org/10.7554/eLife.34970.030

### Data availability

Sequencing data have been deposited in GEO under accession codes GSE108848 and GSE100623.

The following datasets were generated:

| Author(s) | Year | Dataset title | Dataset URL | Database, license, and accessibility information |
|---|---|---|---|---|
| Wang W, Lee S | 2017 | SET-9 and SET-26, the C. elegans homologs of human MLL5, are critical for germline development and longevity | https://www.ncbi.nlm.nih.gov/geo/query/acc.cgi?acc=GSE100623 | Publicly available at the NCBI Gene Expression Omnibus (accession no: GSE100623). |
| Wang W, Lee S | 2017 | SET-9 and SET-26 are H3K4me3 readers and play critical roles in germline development and longevity | https://www.ncbi.nlm.nih.gov/geo/query/acc.cgi?acc=GSE108848 | Publicly available at the NCBI Gene Expression Omnibus (accession no: GSE108848). |

The following previously published dataset was used:

| Author(s) | Year | Dataset title | Dataset URL | Database, license, and accessibility information |
|---|---|---|---|---|
| Lieb J | 2013 | seq-WA30932379_H3K9ac_N2_L3 | http://data.modencode.org/cgi-bin/findFiles.cgi?download=3578 | Available to download at the modENCODE website (accession no: modENCODE_3578) |

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
