## [Decision Letter]

[Editors’ note: a previous version of this study was rejected after peer review, but the authors submitted for reconsideration. The first decision letter after peer review is shown below.]

Thank you for submitting your work entitled "SET-9 and SET-26, the *C. elegans* homologs of human MLL5, are critical for germline development and longevity" for consideration by *eLife*. Your article has been favorably evaluated by a Senior Editor and three reviewers, one of whom served as Guest Reviewing Editor.

Our decision has been reached after consultation between the reviewers. Based on these discussions and the individual reviews below, we regret to inform you that this current version will not be considered further for publication in *eLife*, but would encourage the resubmission of a significantly revised manuscript addressing the concerns of the reviewers.

In the manuscript entitled "SET-9 and SET-26, the *C. elegans* homologs of human MLL5, are critical for germline development and longevity" the authors have done some elegant work to define the overlapping and specific roles of SET-9 and SET-26 regulation of histone methylation in the context of lifespan, stress resistance, and reproduction. Although these are clearly important findings and of interest, the manuscript requires further experiments to validate the claims put forward. Of greatest concern is the confounding effects of the germline defects in the *set-9 set-26* double mutants when comparing ChIPseq and expression profiles to wt animals. The incorporation of tissue specific studies utilizing germlineless worms and dissected gonads would provide clarity to these questions. The concern of the reviewers is that this would take longer than the 2 months afforded by the journal and may result in a story that is much different than the current version. I should note that all reviewers universally recognized the importance of this work and support a new submission of a more developed story to *eLife*.

*Reviewer #1:*

Lee and colleagues present the manuscript, "SET-9 and SET-26, the *C. elegans* homologs of human MLL5, are critical for germline development and longevity" where the overlapping and unique roles of these molecules is explored. This work is an extension of a previous study, by this same group, but provide further details on SET-9 and SET-26 function. Although this study does represent the foundation toward a novel scientific advancement, the manuscript however suffers from a lack of acknowledgement of previous work (some by this group), emphasis of the new findings, and rigor to rule out pleiotropic factors that influence the interpretation of results. That being said, I believe this work could be published in *eLife* in a significantly revised format.

The defects in sterility were not previously associated with mutations in the *set-9; set-26* double mutant. This basis of this phenotype however confounds the interpretation of several of the results of this study, namely:

1) Is the loss of germ cells the direct cause of this stress resistance and longevity response? This was not thoroughly studied in the previous 2012 paper, but the data are in line with previous work that a kri-1 and daf-16-dependent response. To this end, it would be useful to place *set-9* and *set-26* in the context of other established longevity regulators.

2) The authors use heat stress a surrogate for stress resistance, in general, it would be useful to test multiple types of stressors to define the extent to which this pathway impacts survival.

3) Related to the above the function of these K9me3 binding proteins in the context of other established SET proteins is needed.

4) The loss of germ cells, in my opinion, confounds the interpretation of the ChIP and the expression analyses. With a reduction of the germline, it is not surprising that reproduction and developmental genes were identified. The gonad is one of the few tissues that can be readily extracted from the worm. Although, this is a significant undertaking, I believe that it is necessary to dissect (pun intended) the somatic and germ cell specific changes.

The authors should make use of available resources to alter the expression of these targets specifically in the germline or in the soma (expression and RNAi based). This will also help to clarify the new results from above.

The sperm-related changes are intriguing, but bring into question whether the responses observed would also manifest to the same degree in males?

Lastly, the references to MLL5 are distracting in this manuscript and not supported. Unless the authors specifically look at MLL5, which would be amazing, but not essential in my opinion, this should be removed except in the Discussion, where appropriate.

The authors are missing several controls to quantify the degree of expression changes in all strains. This should be straightforward with the reagents made.

*Reviewer #2:*

Wang et al. studied SET-9 and SET-26, two paralogous SET-domain containing genes in *C. elegans* that were previously implicated in lifespan regulation. They characterized various aspects of the phenotypes of single and double mutants and binding profiles of the proteins, concluding that the proteins "organize local chromatin environment to regulate the expression of specific target genes." The paper contains some interesting observations that are a start to understanding what SET-9 and SET-26 do, but their functions and mechanisms of action on chromatin are only vaguely described. Additionally, expression profiling and ChIP seq analyses of the *set-9 set-26* double mutant L4 animals are flawed because the mutant germ line is morphologically very different from wild-type. I do not think that this paper is yet ready for publication.

The beginning part of the paper is generally solid. Using putative null mutants, the authors show that SET-26 alone is important for lifespan and heat stress regulation, but that SET-26 and SET-9 have a redundant function in transgenerational fertility. The mitotic germ cell pool is 3X smaller in *set-9 set-26* double mutants, and 50-70% of mutants are sterile from the F2 stage onward. Consistent with this, they found that SET-9 expression appears restricted to the germ line whereas SET-26 is widely expressed. They also show that the PHD finger of these proteins binds to H3K4me3 when adjacent to acetylated H3K9 or H3K14.

They then performed RNA-seq on wild-type, *set-9* and *set-26* single mutants and *set-9 set-26* double mutants at the L4 stage. They found genes with altered expression in the different mutant L4 strains compared to wild-type and analysed the gene sets. A major problem with this analysis is that the germ lines of *set-9 set-26* mutants are very small compared to wild-type and the mutants have defective gametogenesis. Since the germ line of L4s constitutes a large fraction of the animal, differences in tissue type will confound the comparisons. That is, it is impossible to distinguish gene expression differences that are due to profiling different tissues (*set-9 set-26* double mutants have a lower ratio of germ line to somatic tissue and the germline tissue is morphologically different) from direct effects of *set-9 / set-26* loss. The ChIP-seq experiments have a similar defect. Because of this non-matching of samples, the conclusions made from these analyses are unfortunately not supported.

To control for phenotypic differences, the authors could perform RNAseq on hand picked or sorted *set-9 set-26* double mutants from heterozygous mothers. ChIP would be more difficult since more material is needed, but they could consider combining with a temperature sensitive mutant lacking a germ line, so they are profiling soma in all cases. Alternatively, they could profile a stage where the germ line is not a large fraction of the animal (e.g., embryos or L1 larvae). Another option would be to concentrate studies on *set-26* mutants and the lifespan/stress phenotypes, as they are not sterile.

As noted by the authors, previous work showed that mutants with either increased or decreased H3K4me3 (e.g., HMTs or demethylases) have alterations in both lifespan and fertility, with a variety of germline phenotypes documented including mortal germlines and somatic trans-differentiation of the germline observed in different mutants. More analysis of the *set-9 set-26* sterility phenotype and its relationship to these other mutants (e.g., by phenotype comparison and genetic interactions) would help to clarify their roles. Similarly, the relationship of the *set-9 set-26* lifespan phenotype to other genes affecting lifespan would be informative.

*Reviewer #3:*

In the paper entitled "SET-9 and SET-26, the *C. elegans* homologs of human MLL5, are critical for germline development and longevity" Wang W et al. examine the role of two SET domain containing homologues SET-9 and SET-26 in *C. elegans*. They generate a series of tools including knock-outs and tagged transgenic strains to decipher the individual roles of these duplicated proteins biologically, genomically, and epigenomically. They validate some of their earlier findings (using just RNAi) that *set-26* but not *set-9* regulates worm lifespan, and further find that deletion of both genes causes a severe reduction in brood size that is accompanied with smaller germlines, and display a progressive fertility decline. They use tagged transgenic strains to reproduce earlier results published from their group using an antibody, showing that SET-9 is restricted in expression to the germline while *set-26* is expressed in both the soma and the germline presence. They have exciting preliminary data suggesting that *set-26*'s function in the soma is responsible for its extended longevity. They also examine the effect on gene expression of *set-9* and *set-26* deletion as well as the genomic localization of SET-9 and SET-26. They find that SET-9 and SET-26 bind to H3K4me3 regions and advance a hypothesis that these proteins bind to H3K4me3 to prevent the spreading of H3K4me3 into other regions.

In general, I feel this is an interesting and timely study. I felt that this was a more thorough examination of the roles of SET-9 and SET-26 than their earlier work yet still think that there are some additional experiments that are needed to really nail down the role of these proteins. I feel that delving a little deeper using some *C. elegans* tricks to examine things in a more tissue-specific manner would really help strengthen the paper as would a little deeper molecular analysis. The authors also need to discuss their work in the context of what they had previously shown and should not ignore their own published work.

1) This group earlier reported (Ni Z et al. Aging Cell) that SET-26 was an H3K9 trimethylase. The authors need to address this discrepancy with their own work. In this same work they showed that *set-26* deletion had no effect on H3K4me3 levels. Therefore, saying that it was a surprising result that SET-26 was a K9 methyltransferase after they had advanced that conclusion themselves is a bit surprising itself! I think the authors conclusions that SET-26 is not a major K9 methylase in vivo are perfectly appropriate based on their ChIPseq data (although it would be good to show this at least in supplemental) but they should also discuss their own previous work better to help explain why they feel more confident in their new conclusions.

2) What genes are mis-expressed in the germline or soma of *set-9* and *set-26* mutants? Since the worm is so physically different the comparison of whole worm RNAseq is not really that informative (having a smaller germline will obviously reveal mis-expression of reproduction and development genes!). The soma and the germline can be physically dissociated and RNAseq or analysis of specific genes could be performed by qRT-PCR.

3) To compliment the result in Figure 3D it would be nice to specifically knock-down *set-26* in the somatic cells. This would allow more than the deductive reasoning that SET-26 functioning in the soma is required for the phenotype. This could also be complimented by knocking down *set-26* in germlineless worms and showing that it has a larger lifespan extension phenotype than just the elimination of the germline.

4) It seems that the authors major conclusions are that these two SET domain proteins are acting as H3K4me3 binding proteins. The progressive fertility decline is reminiscent of some of the reported fertility defects of set-2 (Robert et al., 2014). It would be interesting to perform epistasis experiments with the set-2 mutants to see if these are functioning in the same pathway.

5) A way to compliment the in vivo binding data for SET-9/SET-26 ChIPseq would be to IP and then perform mass spec on the histones which were IP'd. This would provide a nice complementation. Also it would be a way to directly assess whether the bound histones had acetylation since the modENCODE data is historically not that accurate and the stages are not directly comparable to each other (which should be mentioned in the legends).

---

## [Author Response]

[Editors’ note: the author responses to the first round of peer review follow.]

Reviewer #1:Lee and colleagues present the manuscript, "SET-9 and SET-26, the C. elegans homologs of human MLL5, are critical for germline development and longevity" where the overlapping and unique roles of these molecules is explored. This work is an extension of a previous study, by this same group, but provide further details on SET-9 and SET-26 function. Although this study does represent the foundation toward a novel scientific advancement, the manuscript however suffers from a lack of acknowledgement of previous work (some by this group), emphasis of the new findings, and rigor to rule out pleiotropic factors that influence the interpretation of results. That being said, I believe this work could be published in eLife in a significantly revised format.The defects in sterility were not previously associated with mutations in the set-9; set-26 double mutant. This basis of this phenotype however confounds the interpretation of several of the results of this study, namely:1) Is the loss of germ cells the direct cause of this stress resistance and longevity response? This was not thoroughly studied in the previous 2012 paper, but the data are in line with previous work that a kri-1 and daf-16-dependent response. To this end, it would be useful to place set-9 and set-26 in the context of other established longevity regulators.

We thank the reviewer for pointing this out. Our data indicated that the loss of germline does not explain the stress resistance and longevity phenotypes that we observed. First, the *set-26* single mutant, which is reproductive and has a brood size similar to that of wild-type worms (~10% reduction in brood size, Figure 2A), exhibits significant lifespan extension and heat resistant phenotypes. Second, the *set-9 set-26* double mutant, which has a ~30% reduced brood size in the first generation (Figure 2A), is *not* more resistant to heat stress and does *not* live longer than the *set-26* single mutant. Additional data detailed in the text support the model that somatic SET-26 modulates longevity, and that germline-expressed SET-26 and SET-9 collaborate to regulate germline function.

In addition, as the reviewer suggested, in our 2012 paper, we demonstrated that the lifespan extension phenotype associated with loss of *set-26* is largely dependent on the transcription factor DAF-16. We more clearly stated this point in the revised text. We have additionally provided new RNA-seq data to reveal the transcriptional targets that are regulated by SET-26 in a DAF-16-dependent manner (Figure 4), which are highly enriched for lifespan regulators (Figure 4). We believe these data further support the functional relationship between SET-26 and DAF-16. Additional genetic analyses of *set-26* in the context of other longevity genes will be interesting but is beyond the scope of the current study.

2) The authors use heat stress a surrogate for stress resistance, in general, it would be useful to test multiple types of stressors to define the extent to which this pathway impacts survival.

We thank the reviewer for the suggestion. The idea of testing multiple stressors is interesting but is not the major focus of this study. Our paper focuses on characterizing the roles of SET-26 and SET-9 in longevity and in germline function, and a large amount of data are presented to support our conclusions.

3) Related to the above the function of these K9me3 binding proteins in the context of other established SET proteins is needed.

We believe the reviewer meant that we should test the genetic relationships of *set-9* and *set-26* with other genes encoding SET domain proteins. To identify chromatin regulators that may function with SET-9 and SET-26 in germline regulation, we performed a targeted RNAi screen, testing genes encoding histone methyltransferases (containing SET domains) and histone demethylases, to look for genes that either enhance or repress the moderate fertility phenotype of the *set-9 set-26* mutant at the F1 generation. Interestingly, we found that RNAi knockdown of the components of the MLL complex, which is established to deposit H3K4me3, significantly worsen the fertility defects of the F1 *set-9 set26* mutant worms. Furthermore, we constructed triple mutants lacking *set-9, set-26*, and carrying a partial loss-of-function allele of *set-2*, the methyltransferase component of the MLL complex. Although the partial loss-of-function mutation of *set-2* does not impact fertility on its own, it greatly worsened the fertility defect of the *set-9 set-26* F1 mutant worms (Figure 9). These results suggested that SET-2 and SET-9 and SET-26 act cooperatively to maintain germline function.

4) The loss of germ cells, in my opinion, confounds the interpretation of the ChIP and the expression analyses. With a reduction of the germline, it is not surprising that reproduction and developmental genes were identified. The gonad is one of the few tissues that can be readily extracted from the worm. Although, this is a significant undertaking, I believe that it is necessary to dissect (pun intended) the somatic and germ cell specific changes.

We thank the reviewer for this comment, which was echoed by reviewers 2 and 3. We have re-designed the experiments and generated completely new data to address this concern.

For expression analysis, we first performed RNA-seq using the F1 *set-9 set-26* mutant (hand-picked homozygous mutant progeny from the heterozygous *set-9(rw5) set-26(tm2467)* /nT1 balanced strain), which has a mild brood size phenotype, and a visibly normal germline (Figure 2A). (This experimental design was also suggested by reviewer 2.) The data of this RNA-seq analysis are now reported in Figure 5 of the revised paper. We further compared the gene expression differences between F1 *set-9 set-26* and wild-type to those of F3 *set-9 set-26* and wild-type (data reported in the initial submission, now Figure 5—figure supplement 1) and note that there is a high correlation between the two data sets. Moreover, GO analysis revealed that the GO terms associated with genes showing significant expression change in F1 and F3 *set-9 set-26* double mutants (compared to wild-type) largely overlap.

We additionally performed another set of RNA-seq analysis, in which we compared germlineless worms with or without *set-26*, in order to investigate the SET-26-regulated genes in the soma. Since our genetic data support a role of somatic SET-26 in lifespan modulation, we believed this data set would reveal the SET-26-regulated genes that are likely to modulate longevity. Indeed, GO term analysis revealed functional groups relevant to aging in this data set (Figure 4 of the revised paper).

Lastly, we also performed H3K4me3 ChIP-seq using fertile F1 *set-9 set-26* mutant. Using these new data, we again observed a significant expansion of the H3K4me3 marking surrounding SET-9/SET-26bound regions in the double mutant (Figure 8 of the revised paper). This finding is similar to the observation we made using the partially sterile F3 *set-9 set-26* mutant. Further comparison indicated that the expansion of H3K4me3 marked regions appeared even more noticeable in the F3 double mutant (Figure 8).

The authors should make use of available resources to alter the expression of these targets specifically in the germline or in the soma (expression and RNAi based). This will also help to clarify the new results from above.

The reviewer’s suggestion is an interesting future direction but is outside of the scope of the current study.

The sperm-related changes are intriguing, but bring into question whether the responses observed would also manifest to the same degree in males?

The reviewer raised an interesting point but investigating males is outside of the scope of the current study. We do note that upon careful analyses of the new ChIP-seq and RNA-seq data using the fertile F1 *set-9 set-26* double mutant, we identified a significant enrichment of germline-specific genes among our gene sets, and both sperm- and oocyte-specific genes are evenly represented among them (Figure 5D, Figure 8—figure supplement 2 in revised paper).

Lastly, the references to MLL5 are distracting in this manuscript and not supported. Unless the authors specifically look at MLL5, which would be amazing, but not essential in my opinion, this should be removed except in the Discussion, where appropriate.

We have reduced the emphasis on MLL5 in the revised paper and now only discuss the possibility of MLL5 being a functional homolog of SET-9 and SET-26 in the Discussion.

The authors are missing several controls to quantify the degree of expression changes in all strains. This should be straightforward with the reagents made.

The reviewer might be referring to controls for SET-26 expression change in the *rrf-1* mutant experiment. Appropriate controls are now included to indicate an efficient knockdown of SET-26 expression upon *set-9/-26* RNAi in the *rrf-1* mutant (Figure 3—figure supplement 1D).

In addition, quantitative PCR validation of a set of randomly selected genes that show RNA expression change from the RNA-seq analyses are also included in Table S6 in Supplementary file 1 of the revised paper.

Reviewer #2:Wang et al. studied SET-9 and SET-26, two paralogous SET-domain containing genes in C. elegans that were previously implicated in lifespan regulation. They characterized various aspects of the phenotypes of single and double mutants and binding profiles of the proteins, concluding that the proteins "organize local chromatin environment to regulate the expression of specific target genes." The paper contains some interesting observations that are a start to understanding what SET-9 and SET-26 do, but their functions and mechanisms of action on chromatin are only vaguely described. Additionally, expression profiling and ChIP seq analyses of the set-9 set-26 double mutant L4 animals are flawed because the mutant germ line is morphologically very different from wild-type. I do not think that this paper is yet ready for publication.The beginning part of the paper is generally solid. Using putative null mutants, the authors show that SET-26 alone is important for lifespan and heat stress regulation, but that SET-26 and SET-9 have a redundant function in transgenerational fertility. The mitotic germ cell pool is 3X smaller in set-9 set-26 double mutants, and 50-70% of mutants are sterile from the F2 stage onward. Consistent with this, they found that SET-9 expression appears restricted to the germ line whereas SET-26 is widely expressed. They also show that the PHD finger of these proteins binds to H3K4me3 when adjacent to acetylated H3K9 or H3K14.They then performed RNA-seq on wild-type, set-9 and set-26 single mutants and set-9 set-26 double mutants at the L4 stage. They found genes with altered expression in the different mutant L4 strains compared to wild-type and analysed the gene sets. A major problem with this analysis is that the germ lines of set-9 set-26 mutants are very small compared to wild-type and the mutants have defective gametogenesis. Since the germ line of L4s constitutes a large fraction of the animal, differences in tissue type will confound the comparisons. That is, it is impossible to distinguish gene expression differences that are due to profiling different tissues (set-9 set-26 double mutants have a lower ratio of germ line to somatic tissue and the germline tissue is morphologically different) from direct effects of set-9 / set-26 loss. The ChIP-seq experiments have a similar defect. Because of this non-matching of samples, the conclusions made from these analyses are unfortunately not supported.To control for phenotypic differences, the authors could perform RNAseq on hand picked or sorted set-9 set-26 double mutants from heterozygous mothers. ChIP would be more difficult since more material is needed, but they could consider combining with a temperature sensitive mutant lacking a germ line, so they are profiling soma in all cases. Alternatively, they could profile a stage where the germ line is not a large fraction of the animal (e.g., embryos or L1 larvae). Another option would be to concentrate studies on set-26 mutants and the lifespan/stress phenotypes, as they are not sterile.

We thank the reviewer for this comment and suggestion, which was echoed by reviewers 1 and 3. We have re-designed the experiments and generated completely new data to address this concern.

For expression analysis, we first performed RNA-seq using the F1 *set-9 set-26* mutant. As suggested by the reviewer, we hand-picked homozygous mutant progeny from the heterozygous *set-9(rw5) set26(tm2467)* /nT1 balanced strain, which has a mild brood size phenotype, and a visibly normal germline (Figure 2A). The data of this RNA-seq analysis are now reported in Figure 5 of the revised paper. We further compared the gene expression differences between F1 *set-9 set-26* and wild-type to those of F3 *set-9 set-26* and wild-type (data reported in the initial submission, now Figure 5—figure supplement 1) and note that there is a high correlation between the two data sets. Moreover, GO analysis revealed that the GO terms associated with genes showing significant expression change in F1 and F3 *set-9 set-26* double mutants (compared to wild-type) largely overlap.

We additionally performed another set of RNA-seq analysis, in which we compared germlineless worms with or without *set-26*, in order to investigate the SET-26-regulated genes in the soma. Since our genetic data support a role of somatic SET-26 in lifespan modulation, we believed this data set would reveal the SET-26-regulated genes that are likely to modulate longevity. Indeed, GO term analysis revealed functional groups relevant to aging in this data set (Figure 4 of the revised paper).

Lastly, we also performed H3K4me3 ChIP-seq using fertile F1 *set-9 set-26* mutant. Using these new data, we again observed a significant expansion of the H3K4me3 marking surrounding SET-9/SET-26bound regions in the double mutant (Figure 8 of the revised paper). This finding is similar to the observation we made using the partially sterile F3 *set-9 set-26* mutant. Further comparison indicated that the expansion of H3K4me3 marked regions appeared even more noticeable in the F3 double mutant (Figure 8).

As noted by the authors, previous work showed that mutants with either increased or decreased H3K4me3 (e.g., HMTs or demethylases) have alterations in both lifespan and fertility, with a variety of germline phenotypes documented including mortal germlines and somatic trans-differentiation of the germline observed in different mutants. More analysis of the set-9 set-26 sterility phenotype and its relationship to these other mutants (e.g., by phenotype comparison and genetic interactions) would help to clarify their roles. Similarly, the relationship of the set-9 set-26 lifespan phenotype to other genes affecting lifespan would be informative.

To identify chromatin regulators that may function with SET-9 and SET-26 in germline regulation, we performed a targeted RNAi screen, testing genes encoding histone methyltransferases (containing SET domains) and histone demethylases, to look for genes that either enhance or repress the moderate fertility phenotype of the *set-9 set-26* mutant at the F1 generation. Interestingly, we found that RNAi knockdown of the components of the MLL complex, which is established to deposit H3K4me3, significantly worsen the fertility defects of the F1 *set-9 set-26* mutant worms. Furthermore, we constructed triple mutants lacking *set-9, set-26*, and carrying a partial loss-of-function allele of *set-2*, the methyltransferase component of the MLL complex. Although the partial loss-of-function mutation of *set-2* does not impact fertility on its own, it greatly worsened the fertility defect of the *set-9 set-26* F1 mutant worms. These results suggested that SET-2 and SET-9 and SET-26 act cooperatively to maintain germline function.

Moreover, in our 2012 paper, we demonstrated that the lifespan extension phenotype associated with loss of *set-26* is largely dependent on the transcription factor DAF-16. We more clearly stated this point in the revised text. We have additionally provided new RNA-seq data to reveal the transcriptional targets that are regulated by SET-26 in a DAF-16-dependent manner (Figure 4), which are highly enriched for lifespan regulators (Figure 4). We believe these data further establish the functional relationship between SET-26 and DAF-16. Additional genetic analyses of *set-26* in the context of other longevity genes will be interesting but is beyond the scope of the current study.

Reviewer #3:In the paper entitled "SET-9 and SET-26, the C. elegans homologs of human MLL5, are critical for germline development and longevity" Wang W et al. examine the role of two SET domain containing homologues SET-9 and SET-26 in C. elegans. They generate a series of tools including knock-outs and tagged transgenic strains to decipher the individual roles of these duplicated proteins biologically, genomically, and epigenomically. They validate some of their earlier findings (using just RNAi) that set-26 but not set-9 regulates worm lifespan, and further find that deletion of both genes causes a severe reduction in brood size that is accompanied with smaller germlines, and display a progressive fertility decline. They use tagged transgenic strains to reproduce earlier results published from their group using an antibody, showing that SET-9 is restricted in expression to the germline while set-26 is expressed in both the soma and the germline presence. They have exciting preliminary data suggesting that set-26's function in the soma is responsible for its extended longevity. They also examine the effect on gene expression of set-9 and set-26 deletion as well as the genomic localization of SET-9 and SET-26. They find that SET-9 and SET-26 bind to H3K4me3 regions and advance a hypothesis that these proteins bind to H3K4me3 to prevent the spreading of H3K4me3 into other regions.In general, I feel this is an interesting and timely study. I felt that this was a more thorough examination of the roles of SET-9 and SET-26 than their earlier work yet still think that there are some additional experiments that are needed to really nail down the role of these proteins. I feel that delving a little deeper using some C. elegans tricks to examine things in a more tissue-specific manner would really help strengthen the paper as would a little deeper molecular analysis. The authors also need to discuss their work in the context of what they had previously shown and should not ignore their own published work.1) This group earlier reported (Ni Z et al. Aging Cell) that SET-26 was an H3K9 trimethylase. The authors need to address this discrepancy with their own work. In this same work they showed that set-26 deletion had no effect on H3K4me3 levels. Therefore, saying that it was a surprising result that SET-26 was a K9 methyltransferase after they had advanced that conclusion themselves is a bit surprising itself! I think the authors conclusions that SET-26 is not a major K9 methylase in vivo are perfectly appropriate based on their ChIPseq data (although it would be good to show this at least in supplemental) but they should also discuss their own previous work better to help explain why they feel more confident in their new conclusions.

We thank the reviewer for pointing this out and we would like to clarify our previous findings in more details. In our Ni et al. paper (2012), we monitored the total levels of four histone modifications, H3K4me3, H3K36me3, H3K9me3, H3K27me3 at young and old time points in germlineless *glp-1* mutant with or without *set-26*. There, we observed that normalized H3K4me3 levels were not impacted by the loss of *set-26*. For the repressive marks H3K9me3 and H3K27me3, we observed that their normalized levels drastically reduce with aging in the *glp-1* single mutant, but their levels remained largely stable in the *glp-1; set-26* double mutants. The data suggested that loss of *set-26* somehow prevented the age-dependent reduction of these repressive marks in the soma. These data alone did not indicate that SET-26 is the enzyme responsible for depositing either of the repressive marks. Indeed, in our Ni et al. paper (pg. 316, 323), we speculated that SET-9 and SET-26 likely do NOT have methytransferase activity, based on changed amino acids in critical residues known to be required for SET domain enzymatic activity, and also that their putative homologs MLL5 (in mammals) and Upset (in flies) are suggested to lack methyltransferase activity.

Subsequent to our Ni et al. 2012 paper, Greer et al. (2014) reported that the SET domain of SET-26 contains in vitro methyltransferase activity and is able to methylate H3K9me3. This was a surprising result to us, and that is why we investigated both the total levels and the genome-wide profiles of H3K9me3 in wild-type and *set-9 set-26* double mutant worms in this study. We did not observe any differences in the levels or genome-wide distribution of H3K9me3 in worms with or without *set-9* and *set-26*.

In the case of H3K4me3, we first examined the total levels and the genome-wide profiles in whole worms of wild-type and *set-9 set-26* double mutant. We observed elevated levels and expanded marking of H3K4me3, which appeared to be more noticeable in the germline. We further compared the genome-wide profiles of H3K4me3 in the germlineless *glp-1* background with or without *set-26* using ChIP-seq. Consistent with our previous results, we observed no significant changes in the genome-wide pattern of H3K4me3 in the *set-26; glp-1* double mutant compared with *glp-1*. Together, the data suggested that, loss of *set-9* and *set-26*, results in expansion of H3K4me3 marking surrounding regions normally bound by SET-9 and SET-26 only in the germ cells.

2) What genes are mis-expressed in the germline or soma of set-9 and set-26 mutants? Since the worm is so physically different the comparison of whole worm RNAseq is not really that informative (having a smaller germline will obviously reveal mis-expression of reproduction and development genes!). The soma and the germline can be physically dissociated and RNAseq or analysis of specific genes could be performed by qRT-PCR.

We thank the reviewer for this comment, which was echoed by reviewers 1 and 2. We have re-designed the experiments and generated completely new data to address this concern.

For expression analysis, we first performed RNA-seq using the F1 *set-9 set-26* mutant (hand-picked homozygous mutant progeny from the heterozygous *set-9(rw5) set-26(tm2467)* /nT1 balanced strain), which has a mild brood size phenotype, and a visibly normal germline (Figure 2A). (This experimental design was also suggested by reviewer 2.) The data of this RNA-seq analysis are now reported in Figure 5 of the revised paper. We further compared the gene expression differences between F1 *set-9 set-26* and wild-type to those of F3 *set-9 set-26* and wild-type (data reported in the initial submission, now Figure 5—figure supplement 1) and note that there is a high correlation between the two data sets. Moreover, GO analysis revealed that the GO terms associated with genes showing significant expression change in F1 and F3 *set-9 set-26* double mutants (compared to wild-type) largely overlap.

We additionally performed another set of RNA-seq analysis, in which we compared germlineless worms with or without *set-26*, in order to investigate the SET-26-regulated genes in the soma. Since our genetic data support a role of somatic SET-26 in lifespan modulation, we believed this data set would reveal the SET-26-regulated genes that are likely to modulate longevity. Indeed, GO term analysis revealed functional groups relevant to aging in this data set (Figure 4 of the revised paper).

3) To compliment the result in Figure 3D it would be nice to specifically knock-down set-26 in the somatic cells. This would allow more than the deductive reasoning that SET-26 functioning in the soma is required for the phenotype. This could also be complimented by knocking down set-26 in germlineless worms and showing that it has a larger lifespan extension phenotype than just the elimination of the germline.

We thank the reviewer for pointing this out. We previously showed that knockdown of *set-9/-26* in the germlineless *glp-1* mutant extends lifespan. We have clarified this in the revised text accordingly.

4) It seems that the authors major conclusions are that these two SET domain proteins are acting as H3K4me3 binding proteins. The progressive fertility decline is reminiscent of some of the reported fertility defects of set-2 (Robert et al., 2014). It would be interesting to perform epistasis experiments with the set-2 mutants to see if these are functioning in the same pathway.

We thank the reviewer for this suggestion. To identify chromatin regulators that may function with SET-9 and SET-26 in germline regulation, we performed a targeted RNAi screen, testing genes encoding histone methyltransferases (containing SET domains) and histone demethylases, to look for genes that either enhance or repress the moderate fertility phenotype of the *set-9 set-26* mutant at the F1 generation. As predicted by the reviewer, we found that RNAi knockdown of the components of the MLL complex, which is established to deposit H3K4me3, significantly worsen the fertility defects of the F1 *set-9 set-26* mutant worms. Furthermore, we constructed triple mutants lacking *set-9, set-26*, and carrying a partial loss-of-function allele of *set-2*, the methyltransferase component of the MLL complex. Although the partial loss-of-function mutation of *set-2* does not impact fertility on its own, it greatly worsened the fertility defect of the *set-9 set-26* F1 mutant worms. These results suggested that SET-2 and SET-9 and SET-26 act cooperatively to maintain germline function.

5) A way to compliment the in vivo binding data for SET-9/SET-26 ChIPseq would be to IP and then perform mass spec on the histones which were IP'd. This would provide a nice complementation. Also it would be a way to directly assess whether the bound histones had acetylation since the modENCODE data is historically not that accurate and the stages are not directly comparable to each other (which should be mentioned in the legends).

The reviewer’s suggestion on follow-up IP-mass spec analysis will be a worthwhile future endeavor but is beyond the scope of the current study. As suggested by the reviewer, we have included caveats relate to the modENCODE data in the revised paper.